# CHARGE syndrome modeling using patient-iPSCs reveals defective migration of neural crest cells harboring CHD7 mutations

Hironobu Okuno[1], Francois Renault Mihara[1], Shigeki Ohta[1], Kimiko Fukuda[2], Kenji Kurosawa[3], Wado Akamatsu[1,4], Tsukasa Sanosaka[1], Jun Kohyama[1], Kanehiro Hayashi[5], Kazunori Nakajima[5], Takao Takahashi[6], Joanna Wysocka[7,8,9,10], Kenjiro Kosaki[11], Hideyuki Okano[1]*

[1]Department of Physiology, Keio University School of Medicine, Tokyo, Japan; [2]Department of Biological Science, Tokyo Metropolitan University, Tokyo, Japan; [3]Division of Medical Genetics, Kanagawa Children's Medical Center, Yokohama, Japan; [4]Center for Genomic and Regenerative Medicine, Juntendo University School of Medicine, Tokyo, Japan; [5]Department of Anatomy, Keio University School of Medicine, Tokyo, Japan; [6]Department of Pediatrics, Keio University School of Medicine, Tokyo, Japan; [7]Department of Developmental Biology, Stanford University School of Medicine, Stanford, United States; [8]Howard Hughes Medical Institute, Stanford University School of Medicine, Stanford, United States; [9]Institute for Stem Cell Biology and Regenerative Medicine, Stanford University School of Medicine, Stanford, United States; [10]Department of Chemical and Systems Biology, Stanford University School of Medicine, Stanford, United States; [11]Center for Medical Genetics, Keio University School of Medicine, Tokyo, Japan

*For correspondence:
hidokano@a2.keio.jp

**Abstract** CHARGE syndrome is caused by heterozygous mutations in the chromatin remodeler, *CHD7,* and is characterized by a set of malformations that, on clinical grounds, were historically postulated to arise from defects in neural crest formation during embryogenesis. To better delineate neural crest defects in CHARGE syndrome, we generated induced pluripotent stem cells (iPSCs) from two patients with typical syndrome manifestations, and characterized neural crest cells differentiated in vitro from these iPSCs (iPSC-NCCs). We found that expression of genes associated with cell migration was altered in CHARGE iPSC-NCCs compared to control iPSC-NCCs. Consistently, CHARGE iPSC-NCCs showed defective delamination, migration and motility in vitro, and their transplantation *in ovo* revealed overall defective migratory activity in the chick embryo. These results support the historical inference that CHARGE syndrome patients exhibit defects in neural crest migration, and provide the first successful application of patient-derived iPSCs in modeling craniofacial disorders.
DOI: https://doi.org/10.7554/eLife.21114.001

## Introduction

CHARGE syndrome is an autosomal dominant genetic disorder characterized by <u>c</u>oloboma of iridis, <u>h</u>eart defect, <u>a</u>tresia choanae, <u>r</u>etarded growth, <u>g</u>enital hypoplasia, and <u>e</u>ar anomalies, a constellation of non-randomly associated malformations (*Blake and Prasad, 2006*). This syndrome is relatively common, occurring approximately one in 10,000 births. Since the first report that de novo mutations

**eLife digest** CHARGE syndrome is a disease in which organs including the heart, eyes and ears may not develop properly. The cells that form the tissues affected by CHARGE syndrome develop in embryos from precursor cells called neural crest cells. Individuals with CHARGE syndrome also have mutations in a gene called *CHD7*. However, it is difficult to examine how *CHD7* mutations affect neural crest cells in embryos.

In recent years, cell reprogramming techniques have made it possible to create induced pluripotent stem cells (iPSCs) from the specialized somatic cells found in the human body. These iPSCs can be developed into many different cell types, including neural crest cells.

Okuno et al. created iPSCs from the skin cells of people with CHARGE syndrome, developed these cells into neural crest cells, and compared them with neural crest cells that were developed from the skin cells of people without CHARGE syndrome. The neural crest cells developed from people with CHARGE syndrome showed multiple abnormalities. For example, they were not able to move around correctly. This is an important observation because neural crest cells must move through tissues to form the various organs affected by CHARGE syndrome.

Okuno et al. also observed changes in the activity of many genes other than *CHD7* in the neural crest cells developed from CHARGE patients. Further research is now needed to find out which genes are the most important for restoring the normal activity of neural crest cells.
DOI: https://doi.org/10.7554/eLife.21114.002

in *CHD7* (chromodomain helicase DNA binding protein 7) might be the cause of CHARGE syndrome (*Vissers et al., 2004*), several groups have sought to identify genotype-phenotype correlations and to determine how various phenotypic features of CHARGE are contributed to by *CHD7* mutations (*Aramaki et al., 2006a*; *Sanlaville et al., 2006*; *Zentner et al., 2010*).

CHD7 is expressed in various cell types, including pluripotent stem cells and cells of the neural tube and placodal regions (*Aramaki et al., 2007*). CHD7 modulates chromatin formation by binding to genomic DNA and regulating the expression of downstream genes (*Martin, 2010*). For instance, several transcriptional factors, such as SOX2, SOX9, and SOX10, have been reported to cooperate with CHD7 in regulating early development in various cell types (*Bajpai et al., 2010*) (*He et al., 2016*) (*Jones et al., 2015*) (*Micucci et al., 2014*) (*Schnetz et al., 2010*).

The hypothesis that clinical features observed in CHARGE syndrome patients are caused by abnormalities in neural crest development has been proposed for more than 30 years (*Siebert et al., 1985*). The cells of the neural crest contribute to many different tissue lineages, including those of the craniofacial skeleton, cranial nerves (VII, VIII, IX and X), ears, eyes, and heart. Since many of the defects observed in CHARGE syndrome appear to be related to abnormalities of cranial neural crest cells, this syndrome is considered as a 'neurocristopathy' (*Aramaki et al., 2007*; *Sanlaville et al., 2006*; *Siebert et al., 1985*). A recent study supports this view by showing that the knockdown of CHD7 in human embryonic stem cells (hESCs) results in migratory neural crest formation defects (*Bajpai et al., 2010*). Moreover, the knockdown of *Chd7* in *Xenopus laevis* or zebrafish embryos led to abnormalities in neural crest specification and migration (*Asad et al., 2016*) (*Bajpai et al., 2010*). The hypothesis that the neural crest pathophysiology observed in CHARGE syndrome is attributable to NCC defects has not been examined using patient-derived cells due to technical challenges. In addition, the phenotypic aspects of CHARGE patient-derived NCCs with respect to different migratory behaviors have not been examined in detail.

Thus, there is great potential value in the establishment of in vitro models of this syndrome using patient-derived cells for use in the study of CHARGE pathophysiology. In the present study, we generated NCCs from CHARGE syndrome patient-derived iPSCs, established in vitro models of CHARGE syndrome, and observed defective migration in CHARGE NCCs via in vitro and in vivo experiments.

## Results

### Clinical features of enrolled CHARGE patients and generation of patient-derived iPSCs

To gain mechanistic insights into the pathogenesis of CHARGE syndrome, we enrolled two CHARGE patients, designated patient 1 (CH1) and patient 2 (CH2), in the present study in an effort to generate patient-derived iPSCs. Both patients have a heterozygous nonsense mutation in *CHD7*, and exhibited the typical phenotype of CHARGE syndrome (*Figure 1—source data 1*). We collected fibroblasts from these patients and generated 25 and 23 iPSC clones from CH1 and CH2, respectively, following the four-factor protocol first reported by Takahashi and Yamanaka (*Takahashi et al., 2007*). We selected four lines from CH1 (CH1#7, CH1#11, CH1#20, CH1#25) and three lines from CH2 (CH2#1, CH2#16, CH2#19) for further analyses. As shown in *Figure 1A–C*, these iPSC clones showed characteristics of pluripotent stem cells, including a morphology similar to that of human embryonic stem cells (ESCs) (*Figure 1A*), the expression of pluripotent stem cell markers (TRA1-60 and TRA1-81) (*Figure 1B*), and the capacity for teratoma formation (*Figure 1C*). We confirmed that the CH1-iPSCs and CH2-iPSCs retained the *CHD7* mutations observed in the human dermal fibroblasts (HDFs) of origin, whereas none of the control iPSCs harbored mutations in *CHD7* (*Figure 1D* and *Figure 1—figure supplement 1*). These patient-derived iPSCs enabled us to conduct further in vitro characterization of phenotypes relevant to CHARGE syndrome. Interestingly, the expression levels of *CHD7* mRNA in the iPSCs derived from both patients were significantly lower than those in control iPSCs (*Figure 1E*).

### Generation and characterization of control and CHARGE iPSC-NCCs

Neural crest cells are thought to be the primary cells affected in CHARGE syndrome (*Blake and Prasad, 2006*) (*Sanlaville et al., 2006*). We therefore differentiated the patient-derived iPSCs into NCCs (iPSC-NCCs) using two protocols adapted from previous studies (*Bajpai et al., 2010*) (*Lee et al., 2009*), which we refer to as Methods A and B, respectively (*Figure 2A and D*). As shown in *Figure 2B and E*, control and CHARGE iPSC-NCCs obtained using each method displayed similar morphological features and were indistinguishable at the colony level. We initially examined the expression of neural crest markers, including CD271 (NGFR) and CD57 (B3GAT1) (*Tucker et al., 1984*), by flow cytometric analysis. As shown in *Figure 2C and F*, more than 90% of the cells obtained using these two methods from control and CHARGE iPSCs expressed both CD271 and CD57. The ratio of CD271 (+) CD57 (+) cells per total induced cells from CHARGE iPSCs was same as that of control without regard to the NCC differentiation method. We additionally performed an immuno-cytochemical analysis using additional NCCs markers, including SOX10 and AP2a, and we found that the iPSC-NCCs expressed these neural crest markers similarly (*Figure 2G*). These results indicate that iPSCs can be differentiated into NCCs, and that the NCC differentiation efficacy of CHARGE iPSCs was similar to that of control iPSCs.

The phenotypes of CHARGE syndrome show clear associations with defects in cranial NCCs (*Siebert et al., 1985*) (*Blake et al., 2008*), which prompted us to investigate the expression of OTX2, a cranial marker (*Millet et al., 1996*), by immunocytochemistry. We determined that OTX2 was expressed in both control and CHARGE iPSC-NCCs (*Figure 2H*). Moreover, using *ad hoc* protocols, we were able to differentiate CHARGE iPSC-NCCs into adipocytes, chondrocytes, osteocytes, myofibroblasts, and peripheral neurons (*Figure 2I, J and K*), indicating that these cells exhibit a potential for differentiation similar to control iPSC-NCCs (*Bajpai et al., 2010*) (*Lee and Studer, 2010*). This suggests that mutations in *CHD7* do not affect NCC differentiation per se. These results also indicate that iPSCs from CHARGE patients can be efficiently differentiated into NCCs that express precise positional markers of cranial region and retain the ability to differentiate into cranial neural crest cells, potentially enabling the generation of disease-relevant cellular models of neurocristopathies, such as CHARGE syndrome.

### Transcriptomic differences between control iPSC-NCCs and CHARGE iPSC-NCCs suggest abnormal migration of CHARGE iPSC-NCCs

To identify the cellular functions dysregulated in CHARGE iPSC-NCCs, we performed a global gene expression analysis of NCCs derived from control and CHARGE iPSCs using a microarray. As shown

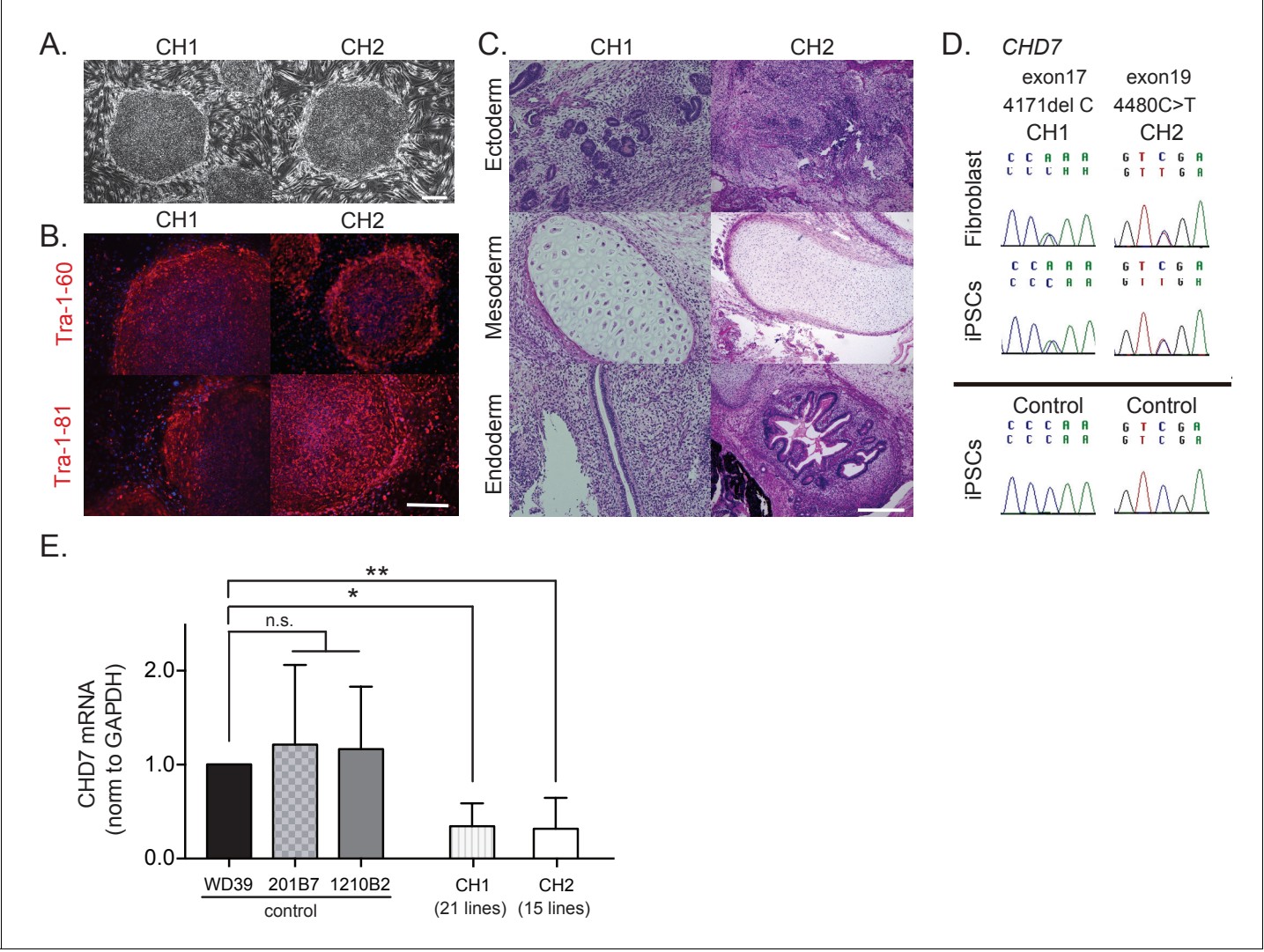

**Figure 1.** Characterization of CHARGE patient-derived iPSCs. (**A**) Representative images of generated iPSCs from CHARGE patients CH1 and CH2 showing human ESC-like morphologies. Bar: 300 μm. (**B**) TRA1-60 and TRA1-81 protein, pluripotent markers, expression in CHARGE iPSCs. Bar: 300 μm. (**C**) Hematoxylin and eosin staining of teratoma derived from CHARGE iPSCs, which were transplanted into the testes of NOD-SCID mice. Bar: 300 μm. (**D**) Direct sequencing analysis of the *CHD7* mutations in CHARGE patient's fibroblasts and iPSCs. The original *CHD7* mutations in the patient's fibroblasts were conserved in the generated iPSCs. The corresponding sequences in control iPSCs are shown below. (**E**) qRT-PCR analysis showed that *CHD7* mRNA expression is significantly reduced in both sets of CHARGE-iPSCs. Control iPSCs (WD39, 201B7, and 1210B2); CH1, 21 lines (CH1#1, #3–17, #19–21, #24, #25); CH2, 15 lines (CH2#1–3, #5, #7–8, #10, #16–23). Technical replicates = 3, Biological replicates (the number of independent sample collection from cells per group) >3, mean ± S.D., n.s.: not significant, *p<0.05, **p<0.01 (Dunn's multiple comparisons test: compared with WD39). The following file is avail able for *Figure 1*,*Figure 1—figure supplement 1*, *Figure 1—source data 1*.

DOI: https://doi.org/10.7554/eLife.21114.003

The following source data and figure supplement are available for figure 1:

**Source data 1.** Features and phenotypes of the enrolled CHARGE patients, and raw data and statistical data of *Figure 1*.

DOI: https://doi.org/10.7554/eLife.21114.005

**Figure supplement 1.** Direct sequencing analysis of the *CHD7* mutations in all iPSC lines used in this study.

DOI: https://doi.org/10.7554/eLife.21114.004

in *Figure 3A*, we found that control and CHARGE iPSC-NCCs expressed essentially similar profiles of marker gene sets for early (*NGFR, B3GAT1, ITGA4*), premigratory (*PAX3, ZIC1*), and migratory (*TWIST1*) NCCs, suggesting the acquisition of the fundamental NCC gene expression profile in CHARGE iPSC-NCCs. The detection of *TWIST1* expression is notable, as one previous study

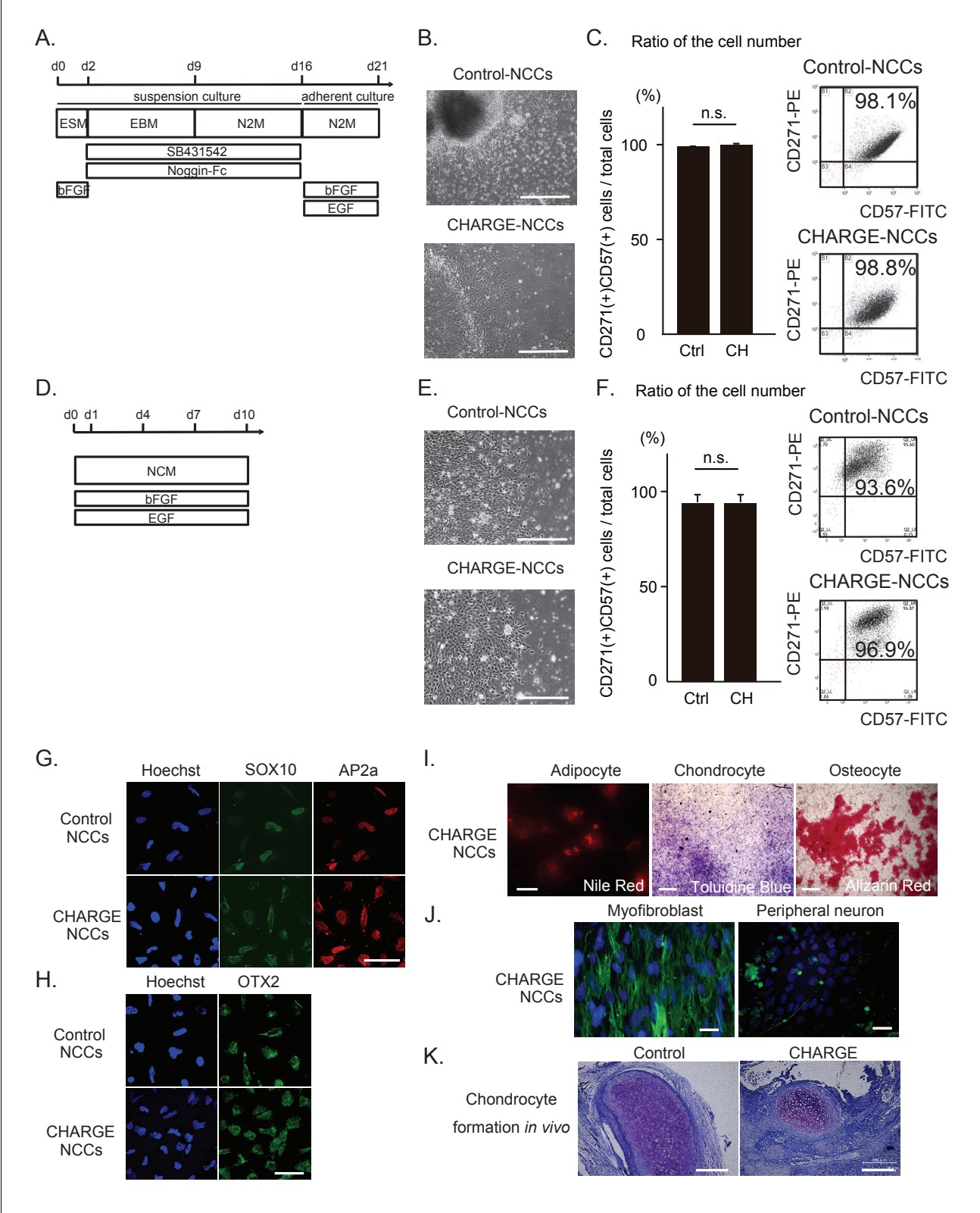

**Figure 2.** Differentiation of patient iPSCs into NCCs. The iPSC-NCCs in (B) and (C) were obtained by Method A, and the iPSC-NCCs in (D)-(K) were obtained by Method B. (A) Schematic presentation of the protocol for NCC differentiation from iPSCs using dual SMAD inhibition (Method A). (B) Representative phase-contrast images of control and CHARGE iPSC-NCCs. Bar: 500 μm. (C) (Left) Quantification of the ratio of the cell number of CD271 and CD57 double-positive cells per total induced cells calculated by flow cytometry. n.s., not significant (Unpaired t test, p=0.77). Biological

*Figure 2 continued on next page*

*Figure 2 continued*

replicates (independent inductions): Control, N = 3; CHARGE, N = 3. (Right) Representative flow cytometric CD271 and CD57 profiles of control and CHARGE iPSC-NCCs. (D) Schematic presentation of the protocol for NCC differentiation from iPSCs through neuroepithelial spheres (Method B). (E) Representative phase-contrast images of control and CHARGE iPSC-NCCs. Bar: 500 µm. (F) (Left) Quantification of the ratio of the cell number of CD271 and CD57 double-positive cells per total induced cells calculated by flow cytometry. n.s., not significant (Unpaired t test, p=0.55). Biological replicates (independent inductions): Control, N = 6; CHARGE, N = 4. (Right) Representative flow cytometric CD271 and CD57 profiles of control and CHARGE iPSC-NCCs. (G) Expression of SOX10 and AP2a protein in control and CHARGE iPSC-NCCs. Bar: 50 µm. (H) Expression of OTX2 in control and CHARGE iPSC-NCCs. Bar: 50 µm. (I) Representative images of CHARGE iPSC-NCCs differentiated into adipocytes, chondrocytes, and osteocytes in vitro. Bars: adipocytes: 50 µm, chondrocytes: 1000 µm, osteocytes: 500 µm. (J) Representative images of CHARGE iPSC-NCCs differentiated into myofibroblasts and peripheral neurons in vitro. Bars: 50 µm. (K) Representative images of control and CHARGE iPSC-NCCs differentiated into chondrocytes by transplantation into the testes of NOD-SCID mice. Bars: 300 µm. The following file is available for *Figure 2C and F*, *Figure 2—source data 1*.

DOI: https://doi.org/10.7554/eLife.21114.006

The following source data is available for figure 2:

**Source data 1.** Raw data and statistical data of *Figure 2*.

DOI: https://doi.org/10.7554/eLife.21114.007

reported that this gene, a marker of migratory NCCs, was downregulated in *CHD7* shRNA-infected human ESC-NCCs compared with control shRNA-infected cells (*Bajpai et al., 2010*).

We next extracted 338 differentially expressed genes (DEGs) (238 upregulated and 100 downregulated in CHARGE iPSC-NCCs) between control and CHARGE iPSC-NCCs (fold change [FC]>1.25) in an effort to identify features common to genes with altered expression in CHARGE iPSC-NCCs (*Figure 3B*). A Gene Ontology (GO) analysis of this set showed enrichment for genes involved in vasculature development (p=2.08E-11), blood vessel development (p=1.24E-11), and blood vessel morphogenesis (p=1.23E-09). Interestingly, GO terms associated with 'cell migration' and 'cell motion' were also significantly enriched in these genes (*Figure 3C*). Given that NCC dysfunction is thought to be linked to the pathogenesis of CHARGE syndrome, we sought to examine the expression of genes associated with NCC behavior, specifically those related to cell migration and cell adhesion. We selected a set of genes listed under each GO term and compared their expression levels in control and CHARGE iPSC-NCCs. As shown in *Figures 3D, E* 56 genes under the GO terms 'migration' or 'adhesion' were differentially expressed. Quantitative real-time PCR (qRT-PCR) analyses performed for four selected genes, *POU3F2*, *OLFM3*, *CTGF* and *EDN1*, confirmed that the changes observed in the microarray data set were indeed significant, thereby providing a validation of the analysis (*Figure 3—figure supplement 1A*). These genes are also listed as CHD7 targets in the dataset. We also performed chromatin immunoprecipitation (ChIP)-qPCR for CHD7 using promoter primers for the genes to reveal direct CHD7 binding to these genes, and we found the direct binding of CHD7 to the distal promoter region in *EDN1* (*Figure 3—figure supplement 1B*). The results of these transcriptome analyses support the notion that NCCs exhibit migratory and/or cell adhesion defects during embryonic development in CHARGE patients.

In the early stages of cranial neural crest cell migration, epithelial-to-mesenchymal transition (EMT) is thought to occur immediately prior to the delamination of NCCs from the neural crest. Migratory NCCs then begin their directed migration along the dorsolateral pathway, reaching their target and initiating their differentiation toward mature cell types (*Kulesa et al., 2010*). Since CHARGE iPSC-NCCs showed only minimal defects in the initiation of NCC differentiation and subsequent differentiation into NCC derivatives, we hypothesized that sequential developmental processes, including delamination and migration, might be disrupted in CHARGE syndrome, as suggested by the results of our transcriptome analysis. We therefore next focused on the dysregulation of CHARGE iPSC-NCCs in cellular adhesion, migration, and cellular motion.

## Defective scattering of CHARGE iPSC-NCCs

The first step of the developmental journey of NCCs consists in their delamination from the region between the dorsal neural tube and the overlying ectoderm (*Kulesa et al., 2010*). We first examined how control iPSC-NCCs migrated outward from spheres using iPSC-NCCs differentiated by Method A to model this particular event. As shown in *Figure 4—video 1*, the cells began to spread out as a continuous monolayer (Phase 1) once the sphere became attached to the culture dish. The control

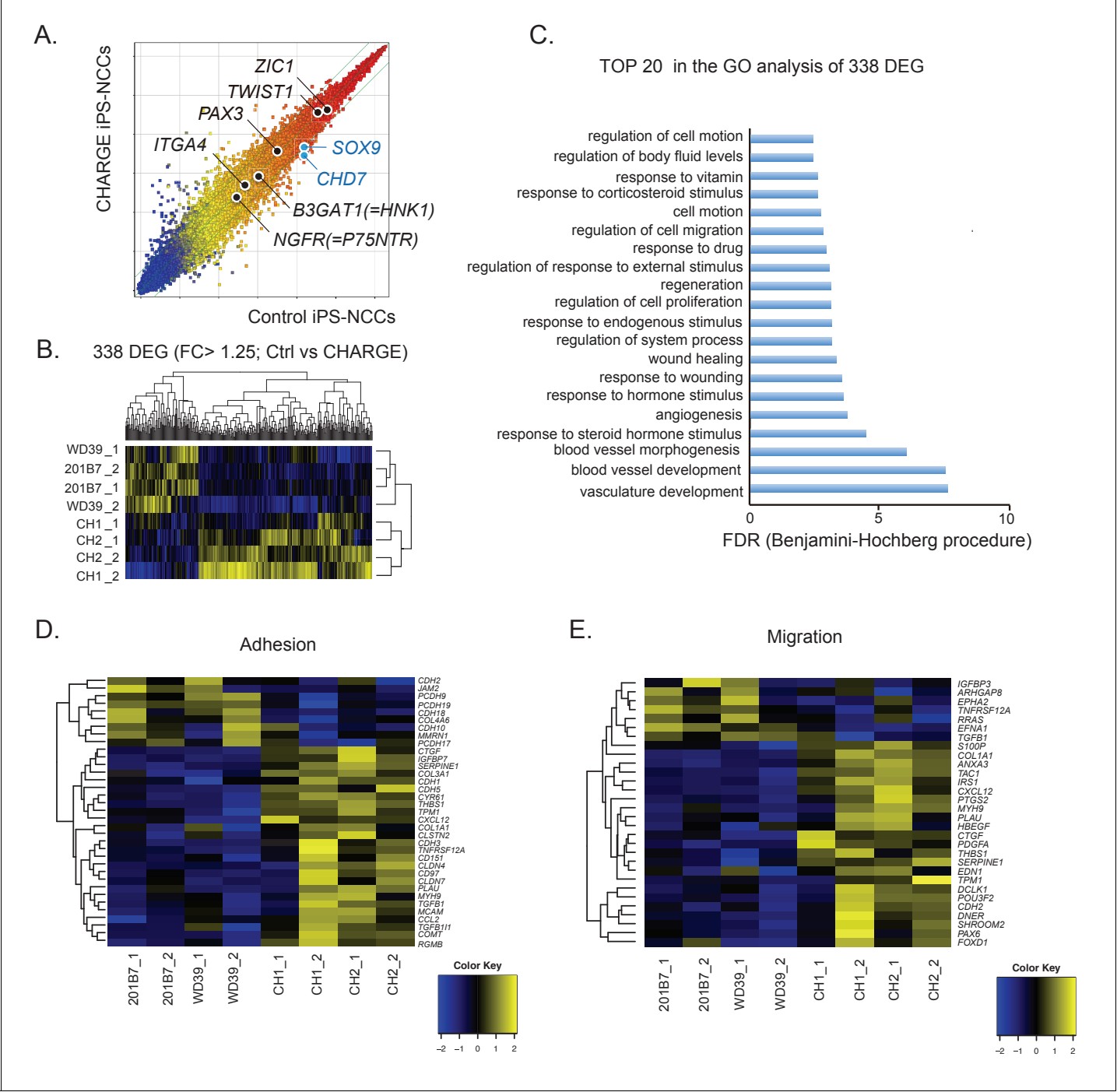

**Figure 3.** Comparative gene expression analysis suggests migration defects in CHARGE iPSC-NCCs. (**A**) Scatter plot of control vs. CHARGE iPSC-NCCs obtained by Method B. Control: 201B7 iPSC-NCCs; CHARGE: CH2#16 iPSC-NCCs. (**B**) Hierarchical clustering of 338 differentially expressed genes (FC >1.25) between control and CHARGE iPSC-NCCs. The control lines consisted of 201B7 and WD39 iPSC-NCCs. The CHARGE lines consisted of CH1#25 (CH1) and CH2#16 (CH2). Each sample from the same line was prepared by independent NCC induction. (**C**) The top 20 list of the GO analysis of the 338 differentially expressed genes between control and CHARGE iPSC-NCCs. (**D**) Hierarchical clustering with 35 genes related to the GO term 'adhesion.' (**E**) Hierarchical clustering with 30 genes related to the GO term 'migration.' The following file is available for *Figure 3*, *Figure 3—figure supplement 1* and *Figure 3—figure supplement 1—source data 1*.

DOI: https://doi.org/10.7554/eLife.21114.008

The following source data and figure supplements are available for figure 3:

**Figure supplement 1.** Examples of differentially expressed genes between control and CHARGE iPSC-NCCs.

*Figure 3 continued on next page*

*Figure 3 continued*

DOI: https://doi.org/10.7554/eLife.21114.009

**Figure supplement 1—source data 1.** Statistical data of *Figure 3—figure supplement 1*

DOI: https://doi.org/10.7554/eLife.21114.010

iPSC-NCCs residing at the outermost periphery then began to scatter apart (Phase 2) (*Figure 4A–a*). In contrast, CHARGE iPSC-NCCs exhibited a distinct behavior in Phase 2, remaining closely associated with their neighbors (*Figure 4A–b*). To clarify this difference, we performed a time-lapse analysis of the initial phase of cell dispersion from the sphere. We used a method to calculate cell dispersion by a Delaunay triangulation algorithm (see Materials and methods) (*Figure 4B*). The distribution of the formed triangular area by the algorithm at 8 hr after the sphere attached to the plate was significantly increased relative to that at t = 0 for both control and CHARGE iPSC-NCCs (*Figure 4C*). Next, we analyzed differences in the increased cell dispersion (from t = 0 to t = 8 hr) between control and CHARGE cells by calculating the median size of the triangular area. We revealed a delayed dispersion of cells from CHARGE spheres (*Figure 4D*). Moreover, to determine whether CHARGE iPSC-NCCs have defects in premigratory-to-migratory transition, we analyzed the intercellular contacts of both control and CHARGE iPSC-NCCs at Phase 2. We visualized the cell associations by F-actin and nuclear staining (*Figure 4E*). Quantitative analysis of the number of intercellular contacts among the outermost migrating cells revealed significantly persistent intercellular contacts in CHARGE iPSC-NCCs in vitro (*Figure 4F*). CHARGE NCCs were reluctant to disperse as single cells, in contrast with control NCCs. This suggests that NCC delamination from the neural tube may be affected in CHARGE syndrome.

## Migratory disabilities in CHARGE iPSC-NCCs

Following delamination from the neural tube, NCCs travel throughout the developing embryo and contribute to major NCC-derived organs (*Cordero et al., 2011*) (*Steventon et al., 2014*) (*Blake et al., 2008*). To assess the migratory ability of CHARGE iPSC-NCCs, we assessed the trans-well migration of dissociated iPSC-NCCs using the xCELLigence system (Roche). Using this system, cells migrating from the upper to lower well through fibronectin-coated microelectrode sensors were monitored automatically (*Figure 5A*). As shown in *Figure 5B*, the migration index of CHARGE iPSC-NCCs became lower than that of control cells after approximately 8 hr of monitoring. At 20 hr, we observed a decrease of approximately 50% in the migration index of the CHARGE iPSC-NCCs compared with that of control cells (*Figure 5C*). To exclude the possibility that the reduction in the number of migrating CHARGE iPSC-NCCs was due to reduced proliferation, we treated control iPSC-NCCs with an antimitotic, aphidicolin. Aphidicolin treatment did not have a significant effect on the migration index of iPSC-NCCs (*Figure 5—figure supplement 1A*). Moreover, a BrdU incorporation assay indicated that the proliferative capacity of CHARGE iPSC-NCCs was not different from that of control iPSC-NCCs (*Figure 5—figure supplement 1B*). Additionally, to exclude the possibility that lower CHARGE iPSC-NCC adherence to fibronectin caused the reduction in the number of migrating CHARGE iPSC-NCCs in this assay, we performed a cell adhesion assay to fibronectin; we found no differential adherence to fibronectin between control and CHARGE iPSC-NCCs (*Figure 5—figure supplement 1C*). Taken together, these results suggest that CHARGE iPSC-NCCs exhibit aberrant migration, in contrast with their preserved capacity for proliferation and adherence to fibronectin.

## Defective spontaneous motility in CHARGE iPSC-NCCs in vitro

The defective scattering and trans-well migration of CHARGE iPSC-NCCs suggest that the collective migration of NCCs is affected. We wondered whether this might be attributable, at least in part, to an intrinsic motility defect of individual CHARGE iPSC-NCCs. Therefore, we performed a time-lapse analysis to examine the single-cell spontaneous motility of control (201B7) and CHARGE (CH1#25) iPSC-NCCs. To exclude cell-density effects, we analyzed the motility of mixed iPSC-NCCs, i.e., control + CHARGE, within the same well (*Figure 6A*). The tracking of individual iPSC-NCCs revealed that the average velocities progressively increased over the course of the recording period (*Figure 6B*). Notably, at any time interval, the average velocity of the CHARGE iPSC-NCCs was

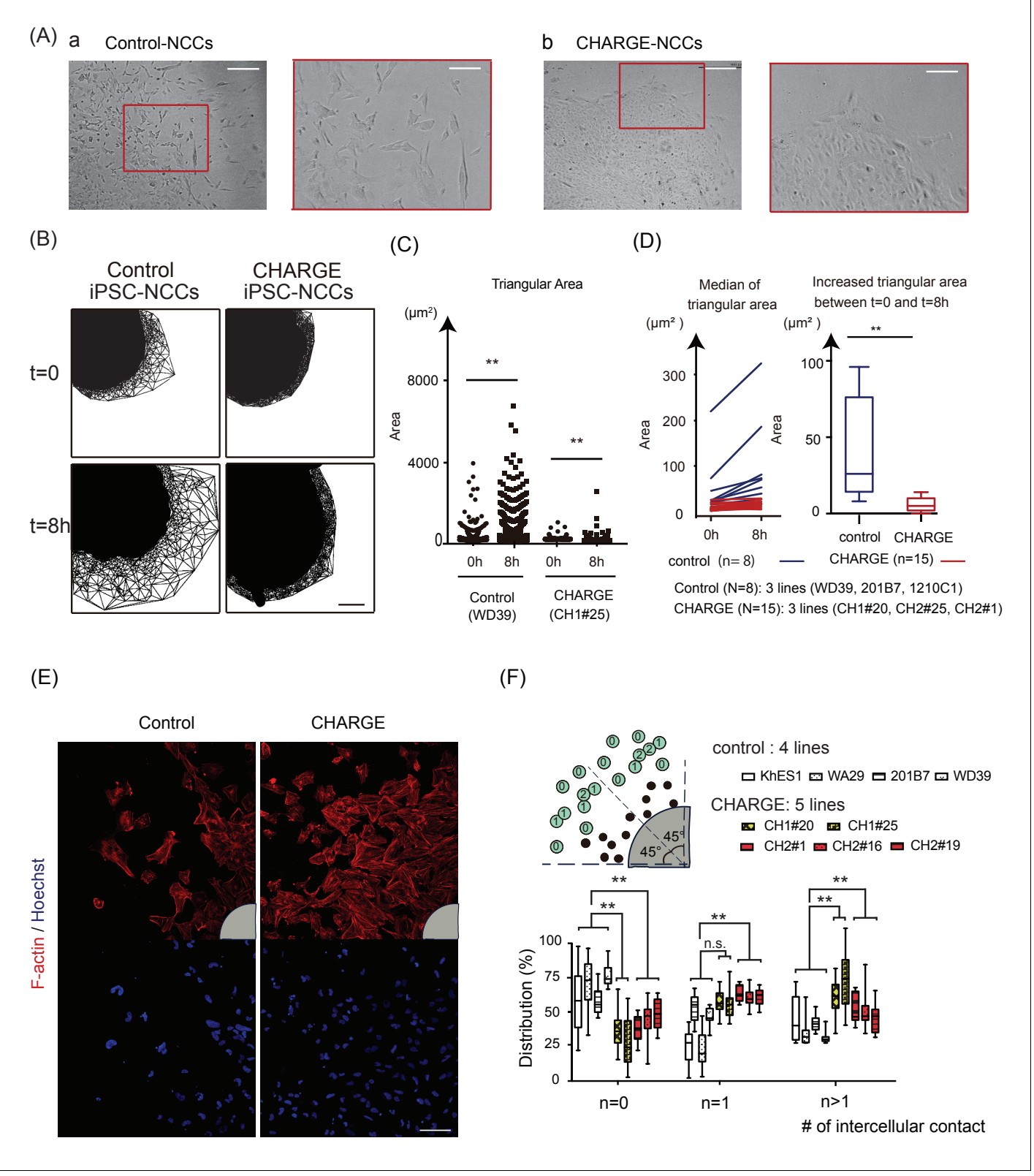

**Figure 4.** Defective Scattering of CHARGE iPSC-NCCs in vitro. (**A**) Representative images of control and CHARGE iPSC-NCCs obtained by Method A. The control iPSC-NCCs residing at the outermost periphery began to scatter apart (**Figure 4A–a**). In contrast, the CHARGE iPSC-NCCs exhibited a distinct behavior in Phase 2, remaining closely associated with their neighbors (**Figure 4A–b**). The red square on the left corresponds to that on the right in both a and b. Bars (left in a and b): 250 μm. Bars (right in a and b): 100 μm. (**B**) Cell dispersion at t = 0, and t = 8 hr was analyzed using the

*Figure 4 continued on next page*

*Figure 4 continued*

Delaunay triangulation algorithm. Control: WD39; CHARGE: CH1#25. Bar:100 µm. (**C**) Dot plots represent the distribution of each triangular area shown in (**B**). (Left) Triangular area of the control iPSC-NCCs at t = 0 and t = 8 hr. **p<0.01 (Mann-Whitney U test, Cohen's d 0.39). (Right) Triangular area of the CHARGE iPSC-NCCs at t = 0 and t = 8 hr. **p<0.01 (Mann-Whitney U test, Cohen's d 0.03). (**D**) Blue bars represent for control iPSC-NCCs, and red bars represent for CHARGE iPSC-NCCs. (Left) Median of the triangular area of the control and CHARGE iPSC-NCCs at t = 0 and t = 8 hr. Control iPSC-NCCs, N = 8 (WD39, 201B7, 1201C1); CHARGE iPSC-NCCs, N = 15 (CH1#20, CH1#25, CH2#1). (Right) Box plots showing the increased median value of the triangular area from t = 0 to t = 8 hr. **p<0.01 (Mann-Whitney U test, Cohen's d 1.66). (**E**) Representative images of iPSCs-NCCs at the outermost periphery visualized with F-actin and nuclear staining. Bar: 100 µm. Gray quarter circles show the postion of each sphere. (**F**) The outermost nine cells (green circle) in each of the eight 45 degree-sector of a sphere were scored by counting the number of their contacting-neighboring cells. The number in a green circle represents the score. The box plots show the distribution of the number of intercellular contacts among the outermost migrating cells in each line. Biological replicates: control, 15 inductions (KhES1, 3; WD39, 3; 201B7, 5; WA29, 4); CH1, 7 inductions (CH1#20, 3; CH1#25, 4); CH2, 12 inductions (CH2#1, 4; CH2#16, 3; CH2#19, 5). Number of cells scored: control, 3707 cells (KhES1, 1017 cells; WD39, 1197 cells; 201B7, 657 cells; WA29, 836 cells); CH1, 3600 cells (CH1#20, 1989 cells; CH1#25, 1611 cells); CH2, 3213 cells (CH2#1, 738 cells; CH2#16, 1791 cells; CH2#19, 684 cells). n = 0; **p<0.01 (Dunnett's multiple comparisons test; Cohen's d 2.67 (control vs CH1), Cohen's d 1.93 [control vs CH2]). n = 1; n.s.; not significant, **p<0.01 (Dunn's multiple comparisons test; Cohen's d (control vs CH1) 1.02, Cohen's d 1.71 [control vs CH2]. n > 1; **p<0.01 (Dunnett's multiple comparisons test; Cohen's d 2.88 (control vs CH1), Cohen's d 1.54 [control vs CH2]). The following file is available for *Figure 4*, *Figure 4—video 1* and *2*, *Figure 4— source data 1*.

DOI: https://doi.org/10.7554/eLife.21114.011

The following video and source data are available for figure 4:

**Source data 1.** Raw data and statistical data of *Figure 4*.
DOI: https://doi.org/10.7554/eLife.21114.012

**Figure 4—video 1.** Time-lapse movies of attached control and CHARGE iPSCs-NCCs.
DOI: https://doi.org/10.7554/eLife.21114.013

**Figure 4—video 2.** Time-lapse movies of attached control and CHARGE iPSCs-NCCs.
DOI: https://doi.org/10.7554/eLife.21114.014

---

lower than that of control iPSC-NCCs (Sidak's multiple comparisons test after two-way repeated measures ANOVA: pTime <0.001, pCellType <0.001; 201B7, N = 80 cells tracked; CH1#25, N = 97 cells tracked). Sidak's multiple comparisons tests confirmed the significantly reduced velocities of CHARGE iPSC-NCCs at multiple time intervals (*Figure 6B*). In contrast, the directionality of the iPSC-NCCs was constant over time and was similar for both control and CHARGE iPSC-NCCs (*Figure 6B*). A comparison of two different cell lines, WD39 and CH2#16 (*Figure 6—figure supplement 1A–B*) yielded similar results; the average velocity of CHARGE iPSC-NCCs migrating as single cells was significantly reduced compared with that of control iPSC-NCCs (Sidak's multiple comparisons test after two-way repeated measures ANOVA: pTime <0.001, pCellType <0.001; WD39, N = 170 cells tracked; CH2#16, N = 133 cells tracked). Sidak's multiple comparisons tests confirmed the significantly reduced velocities of CHARGE iPSC-NCCs at multiple time intervals. In contrast, the directionality of CHARGE iPSC-NCCs was not different to that of control iPSC-NCCs, indicating that the abnormal migration of CHARGE iPSC-NCCs is due, at least in part, to a defective intrinsic motility.

## Defective migration of CHARGE iPSC-NCCs in chick embryos

To examine whether CHARGE iPSC-NCCs also show defective migration in vivo, we grafted iPSC-NCCs into the dorsal edge of the hindbrain of chick embryos (HH stage 8–10). To compare the migration ability of CHARGE iPSC-NCCs and control cells under identical conditions, we transplanted a mixture of control and CHARGE iPSC-NCCs into the same embryo. To distinguish these cells, the iPSC-NCCs were stained with different lipophilic dyes (Vybrant DiI or DiO) before transplantation (*Figure 7A*). First, to examine the serial migration of the transplanted cells, we transferred the transplanted embryo to a glass-bottomed plate (IWAKI) 6 hr after transplantation and then acquired time-lapse images every 20 min (*Figure 7–video 1*). We tracked 4–14 cells migrating well for both control and CHARGE cells in an embryo (the average # of counted cells per experiment: control, 8.9; CHARGE, 9.0), and we calculated their velocity at each time interval using ImageJ (*Figure 7—figure supplement 1A,B*). In 6 of 9 transplanted embryos, the velocity of CHARGE iPSC-NCCs was significantly less than that of control cells. In the other 3 embryos, there were no differences between control and CHARGE cells. (*Figure 7—figure supplement 1—source data 1* -tab1) Collectively, CHARGE iPSC-NCCs exhibited a lower velocity compared with that of the co-

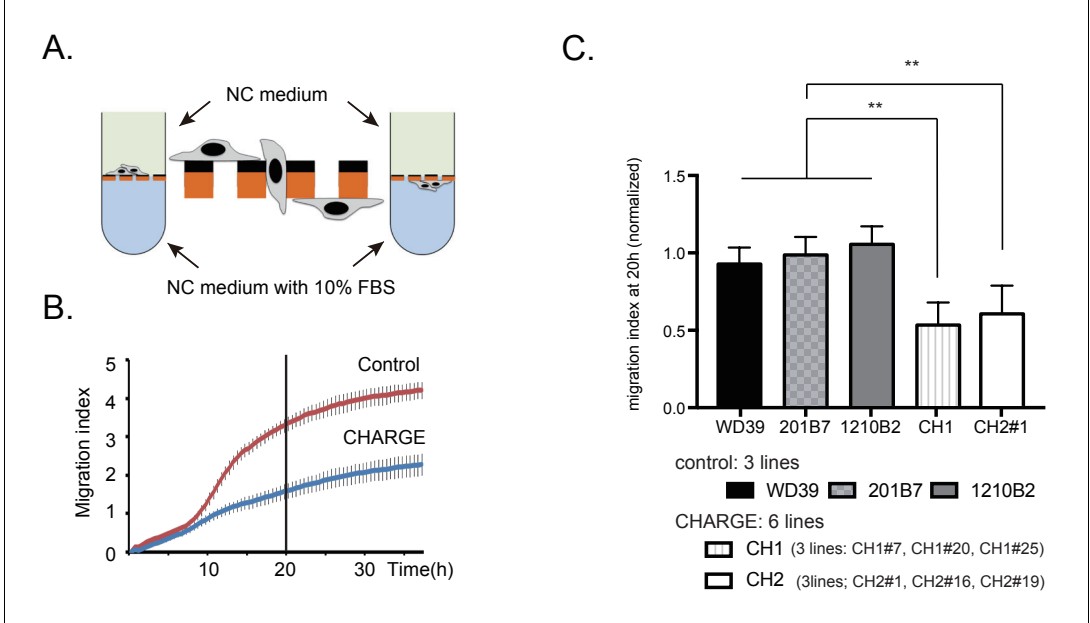

**Figure 5.** Migratory disabilities of CHARGE iPSC-NCCs. (A) xCELLigence. The migration index of iPSC-NCCs was measured using xCELLigence, by which migrating cells through microelectrode sensors were monitored automatically. (B) Representative curve of control (WD39; red) and CHARGE (CH2#1; blue) iPSC-NCCs migration index during 36 hr. Bar: SD. (C) Quantitative analysis of migration index after 20 hr of monitoring with xCELLigence, normalized to the control iPSC-NCCs migration index. Bar: SD. Biological replicates (the number of independent NCC inductions): control, N = 20 (WD39, N = 8; 201B7, N = 7; 1210B2, N = 5); CH1, N = 6 (CH1#7, N = 1, CH1#20, N = 1; CH1#25, N = 4); CH2, N = 11 (CH2#1, N = 5; CH2#16, N = 5; CH2#19, N = 1). Technical replicates: 2–4 per experiment. **$p<0.01$ (Turkey's multiple comparisons test): The following file is available for *Figure 5*, *Figure 5—figure supplement 1*, *Figure 5—source data 1*.

DOI: https://doi.org/10.7554/eLife.21114.015

The following source data and figure supplement are available for figure 5:

**Source data 1.** Raw data of xCelligence assay of iPSC-NCCs in vitro.

DOI: https://doi.org/10.7554/eLife.21114.017

**Figure supplement 1.** Control and CHARGE iPSC-NCCs exhibit similar proliferation and adhesion.

DOI: https://doi.org/10.7554/eLife.21114.016

transplanted control iPSC-NCCs (p=0.03; Wilcoxon signed-rank test) (*Figure 7—figure supplement 1C*). Second, we examined the iPSC-NCCs that had migrated throughout the embryo *in ovo* thirty-six hours after transplantation. Interestingly, the iPSC-NCCs migrated from the site of transplantation (dorsal area) to the ventral area (*Figure 7B*, lower panels). Both control and CHARGE iPSC-NCCs migrated in the expected direction, consistent with the normal developmental routes of NCCs. Notably, CHARGE iPSC-NCCs did not follow abnormal routes to ectopic sites in this model. To compare the migration of control and CHARGE iPSC-NCCs in vivo, we scored the maximum distance that the transplanted cells migrated in 17 surviving chick embryos (*Figure 7—source data 1*). We recorded the locations of the iPSC-NCCs-derived cells that had migrated the greatest distance from the transplant site and assigned a score from 1 (dorsal area) to 4 (ventral area) to each grafted embryo (*Figure 7B*). As shown in *Figure 7C*, CHARGE iPSC-NCCs exhibited similar or lower migration scores compared with those of the co-transplanted control iPSC-NCCs. Adversely, the migration exhibited a large degree of variability among the embryos. These data suggest that the reduced migratory capability of CHARGE iPSC-NCCs observed in vitro reflects their reduced migration in vivo after transplantation in chick embryo.

Our results show that NCCs differentiated from CHARGE iPSCs exhibit migration defects in vitro and in vivo that are consistent with the pathological features of CHARGE syndrome and thus may serve as a useful model for investigating the molecular causes of this condition.

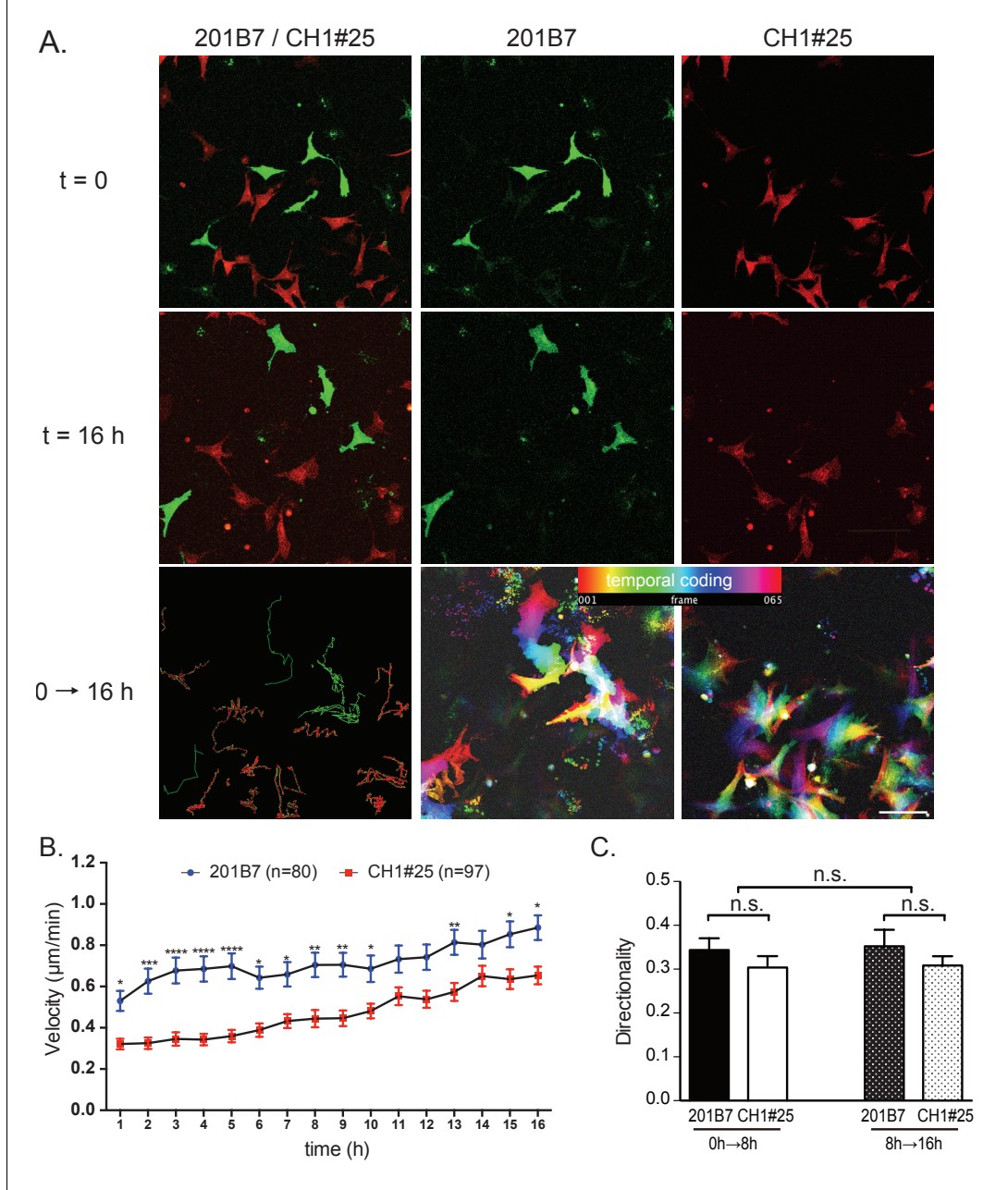

**Figure 6.** Single cell motility analysis of iPSC-NCCs in vitro. (A) Representative images of migratory 201B7 (control; green) and CH1#25 (CHARGE; red) iPSC-NCCs at 0 hr and 16 hr, along with a combined view. Bar: 50 μm. (B) Average velocities of migratory iPSC-NCCs were assessed by tracking control and CHARGE iPSC-NCCs for 16 hr. Number of cells tracked: 201B7, 89 cells tracked; CH1#25, 124 cells tracked. *p<0.05, **p<0.01, ***p<0.001 (Sidak's multiple comparisons tests). (C) Quantitative analysis of the directionality of migratory iPSC-NCCs tracked in (B). n.s., not significant (Tukey's multiple comparisons test). Bars in (B) and (C) display the mean ± SEM. The following file is available for *Figure 6*, *Figure 6—source data 1*, *Figure 6—figure supplement 1*, *Figure 6—figure supplement 1—source data 1*.

DOI: https://doi.org/10.7554/eLife.21114.018

The following source data and figure supplements are available for figure 6:

**Source data 1.** Raw data and statistical data of *Figure 6*

DOI: https://doi.org/10.7554/eLife.21114.021

**Figure supplement 1.** Single-cell motility analysis of iPSC-NCCs using other control and CHARGE iPSC-NCC lines.

DOI: https://doi.org/10.7554/eLife.21114.019

**Figure supplement 1—source data 1.** Raw data and statistical data of *Figure 6—figure supplement 1*.

*Figure 6 continued on next page*

*Figure 6 continued*

DOI: https://doi.org/10.7554/eLife.21114.020

## Discussion

We successfully generated iPSCs from CHARGE syndrome patient-derived fibroblasts and differentiated them into NCCs. We identified multiple functional abnormalities in CHARGE iPSC-NCCs, which may reflect a direct link between the NCC population affected in CHARGE syndrome and the multiple anomalies observed in CHARGE syndrome patients (*Figure 8*). It was previously shown by *CHD7* knockdown in human ESCs that CHD7 controls EMT in multipotent NCCs (*Bajpai et al., 2010*). Our results using CHARGE syndrome patient-derived iPSCs indicate that CHARGE iPSC-NCCs have migratory defects and that a series of migration-related behaviors following EMT, namely, delamination, migration, and motility, are affected.

First, our scattering assay using migratory iPSC-NCCs, which is an in vitro model of the premigratory-to-migratory transition, akin to delamination in vivo, indicated defective delamination in CHARGE NCCs (*Figure 4*). In this assay, iPSCs were induced into neuroectodermal spheres (*Lee et al., 2010*), and the cells migrated out from the spheres. This migration of iPSC-NCCs out from neuroectodermal spheres resembles the premigratory-to-migratory transition, after which the migrating cells scattered as single cells in a manner similar to delamination in vivo. Our finding of defective CHARGE iPSC-NCCs scattering is compatible with a previous report that CHD7 controls the transcriptional reprogramming of EMT. As shown in *Figure 3E*, *FOXD1* expression was upregulated in CHARGE iPSC-NCCs. Since *FOXD1* is known to be expressed in premigratory NCCs and extinguished once migration occurs (*Gómez-Skarmeta et al., 1999*), altered *FOXD1* expression may lead to a defective premigratory-to-migratory transition occurring in CHARGE iPSC-NCCs. While delamination in vivo is not so simple as this in vitro model, as it is subject to complex orchestration by various signals, this delamination model may be a very valuable tool, since it is impractical for ethical and technical reasons to observe human NCC delamination in early embryos directly.

Second, our transwell migration assay using dissociated iPSC-NCCs, which occurred after delamination in vivo, showed defective CHARGE cell migration (*Figure 5*). As shown in *Figure 3D–E*, many genes referred to under the GO terms 'migration' and 'adhesion' were differentially expressed in CHARGE iPSC-NCCs, and the defective migratory phenotype of CHARGE iPSC-NCCs in the transwell assay is compatible with the results of this transcriptional analysis. This assay models cell migration toward chemoattractants. All cranial NCCs are suggested to have similar migratory potential, unlike trunk NCCs, which are known to be a heterozygous population consisting of cells such as leader cells and follower cells (*Richardson et al., 2016*). Therefore, this transwell migration assay is adequate for assessing the migration of cranial NCCs such as our iPSC-NCCs that robustly express OTX2 (*Figure 2H*). Of course, during the long journey from the dorsal neural tube to the ventral area, many signals influence NCC migration in a complex manner, and NCCs change their character during their migration. In our model, the in vivo migration provided additional evidence supporting the defective migration of CHARGE iPSC-NCCs.

Third, a spontaneous motility assay allowed us to assess whether defective motility is a partial cause of the defective migration of CHARGE iPSC-NCCs (*Figure 6*). In this assay with a mixed population of control and CHARGE iPSC-NCCs (co-culture system), autocrine or paracrine factors would likely diffuse within the wells and affect neighboring cells. The observed spontaneous defective motility of the CHARGE cells suggests that such soluble factors are not involved in the defective migration of CHARGE iPSC-NCCs.

Our transcriptome analysis revealed that genes associated with 'migration' and 'adhesion' were altered in CHARGE iPSC-NCCs. CHD7 is an important chromatin remodeler and may thus play roles in various gene regulatory mechanisms (*Bajpai et al., 2010*) (*He et al., 2016*) (*Jones et al., 2015*) (*Micucci et al., 2014*; *Schnetz et al., 2010*). In particular, we focus on the PAX6 downstream and Hippo/YAP pathways. Importantly, CHD7 is considered to function cooperatively with SOX2 as a molecular partner (*Engelen et al., 2011*), and PAX6 has also been reported to be a functional partner of SOX2 (*Thakurela et al., 2016*). As shown in *Figure 3—figure supplement 1A*, the expression levels of *POU3F2* (*BRN2*) and *OLFM3* (Optimedin) were significantly downregulated in CHARGE compared with control iPSC-NCCs, and these two genes have been reported to be targets of PAX6

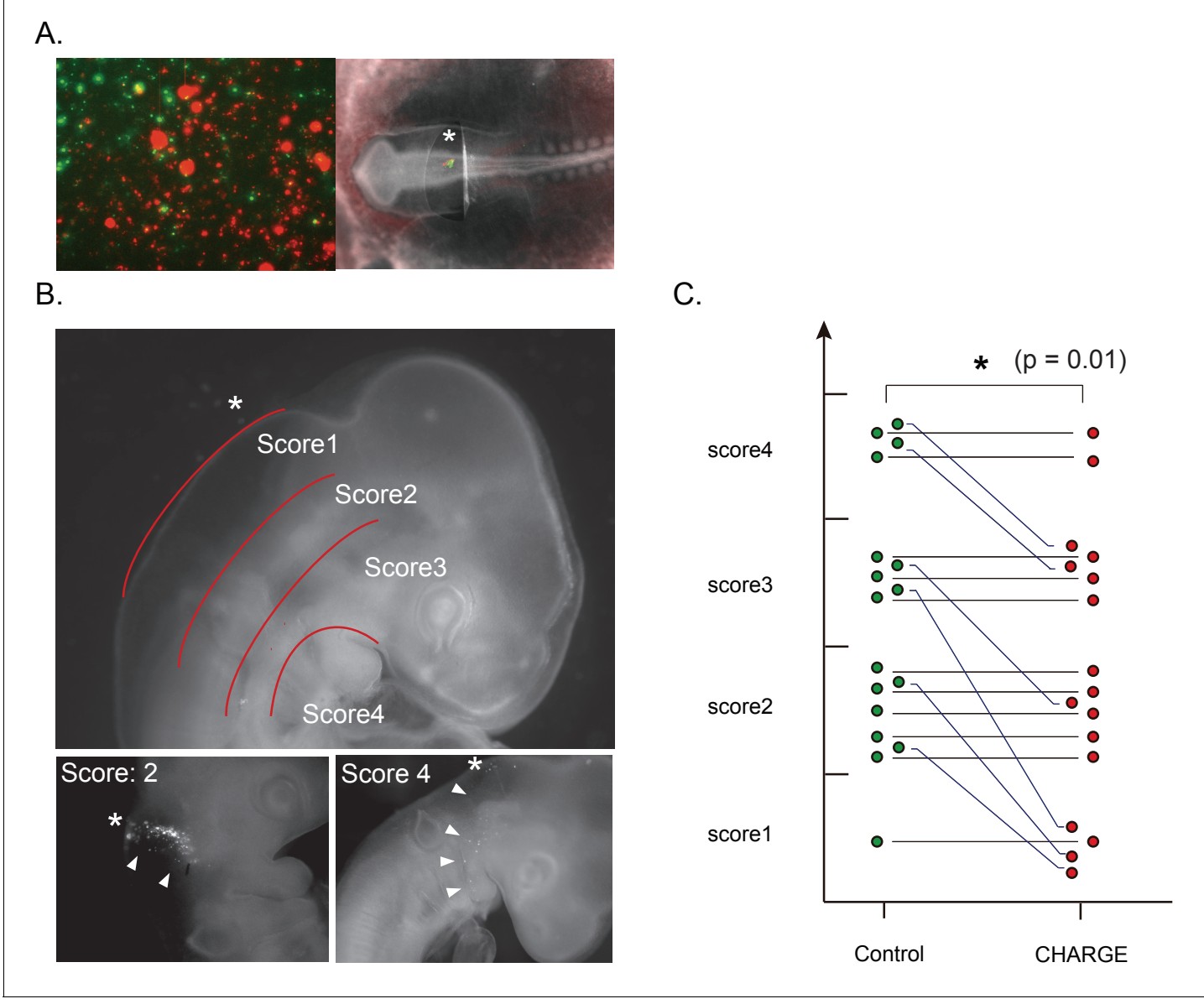

**Figure 7.** Defective migration of CHARGE iPSC-NCCs in chick embryos. (**A**) (Left) Representative image of in vitro control and CHARGE iPSC-NCCs that were prestained with Vybrant DiI and DiO, respectively. (Right) Representative image of chick embryo at the HH 8–10 stage in which iPSC-NCCs were injected around the hindbrain. (**B**) (Upper) Scoring of migration in transplanted chick embryos: Score I, dorsal side of otic cyst; Score II, around otic cyst; Score III, from ventral side of otic cyst to dorsal side of pharyngeal arch; Score IV, migrated into pharyngeal arches. (Lower-left panel) Example of chick embryos with a score of II. (Lower-right panel) Example of chick embryos with a score of IV. The asterisks in the panels of B indicate the NCC transplantation sites. (**C**) Graph of transplanted chick embryo scores; cells transplanted into the same embryo were connected with a line. \*\*p<0.01 (Wilcoxon signed-rank test). The following file is available for *Figure 7*, *Figure 7—figure supplement 1*, *Figure 7—video 1* and *Figure 7—figure supplement 1—source data 1*.

DOI: https://doi.org/10.7554/eLife.21114.022

The following video, source data, and figure supplements are available for figure 7:

**Source data 1.** A list of transplanted cells and scores.
DOI: https://doi.org/10.7554/eLife.21114.025

**Figure supplement 1.** Time-lapse analysis of transplanted NCCs in chick embryos.
DOI: https://doi.org/10.7554/eLife.21114.023

**Figure supplement 1—source data 1.** Raw data of *Figure 7—figure supplement 1*
DOI: https://doi.org/10.7554/eLife.21114.024

**Figure 7—video 1.** A time-lapse movie of transplanted NCCs in chick embryos.

*Figure 7 continued on next page*

*Figure 7 continued*

DOI: https://doi.org/10.7554/eLife.21114.026

(*Grinchuk et al., 2005*) (*Ninkovic et al., 2013*) (*Raviv et al., 2014*). POU3F2 is involved in controlling the migration of melanocytes, which are neural crest derivatives (*Berlin et al., 2012*). OLFM3 is considered to be involved in cell-cell adhesion and cell attachment to the extracellular matrix (*Grinchuk et al., 2005*). Several downstream targets of Pax6 have been identified as cell adhesion molecules and structural proteins (*Cvekl and Callaerts, 2017*). Altered *CHD7* expression resulted in the upregulation of *PAX6* and the downregulation of PAX6 downstream genes (*Figure 3E*). Therefore, it is conceivable that CHD7 regulates multipotent NCC migration by cooperating with PAX6.

Next, *CTGF* and *EDN1*, known to be downstream factors in the Hippo-YAP signaling pathway, are highly expressed in CHARGE iPSC-NCCs compared with control cells (*Figure 3—figure supplement 1A*). The Hippo-YAP signaling pathway is known to be regulated via cell density (*Zhao et al., 2007*), and this pathway has recently been reported to inhibit migration and suggested to play important roles on the early stage of NCC specification and migration (*Lamar et al., 2012*; *Wang et al., 2016*). In particular, CTGF and EDN1 play important roles in craniofacial development, and the timing and regulation of their expression are crucial for their function (*Maj et al., 2016*; *Mercurio et al., 2004*). Altered *CHD7* expression in iPSC-NCCs resulted in the upregulation of *CTGF* and *EDN1*. Therefore, it is conceivable that CHD7 regulates the craniofacial phenotype of CHARGE syndrome through the Hippo-YAP pathway. To clarify this mechanistic insight into how NCCs are dysregulated in CHARGE syndrome patients, it is noteworthy that 202 of the 338 differentially expressed genes between the CHARGE and control iPSC-NCCs were listed as target genes of CHD7 in the ChIP-seq datasets from the ENCODE Transcriptional Factor Target dataset (*Rouillard et al., 2016*). Although these target sites vary depending on cell type, we found the target site of *CHD7* in the EDN1 distal promoter region by ChIP-qPCR for CHD7 using our cells. This result suggests that CHD7 regulates not only the expression of some specific key genes but also the robust gene expression in early NCCs.

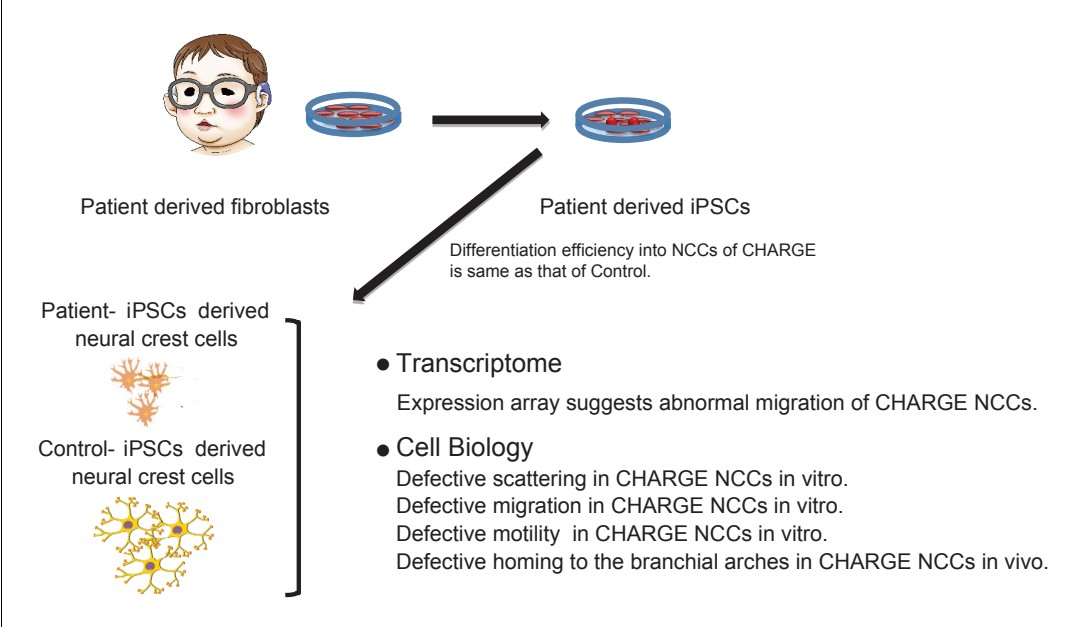

**Figure 8.** Model summarizing defective migration of CHARGE NCCs using patient-derived iPSCs. Defects in cell delamination, migration, and motility in our model reflect phenotypes in CHARGE syndrome that develop *in utero*. Various aspects of NCC migration were not well coordinated in CHARGE NCCs due to the dysfunction of CHD7.

DOI: https://doi.org/10.7554/eLife.21114.027

The current study represents the first model of a developmental morphogenetic disorder using patient-derived iPSCs. To date, the neural crest pathophysiology observed in CHARGE syndrome has not been examined directly using patient-derived cells due to technical challenges and ethical concerns surrounding the collection of NCCs from human embryos. Moreover, since the developmental regulation of NCCs is known to be unique to individual species (*Acloque et al., 2009*; *Barriga et al., 2015*), NCCs derived from CHARGE patient-derived iPSCs are an appropriate source for modeling the cellular features of this disease in vitro.

We suggest that such cells may be used as a powerful assay system for evaluating NCC dysfunction in other morphogenetic disorders that could be considered neurocristopathies, such as craniofacial syndrome (*Minoux and Rijli, 2010*) and infants of vitamin A exposure(*Kraft et al., 1989*; *Rosa, 1983*). NCCs play important roles in the formation of sensory organs, such as ears, eyes, and olfactory organs, and some congenital neurocristopathies are caused by reproductive toxicity. These deformities of experimental animals have been used for the toxicity testing of newly developed drugs. The iPSC-NCC system presented herein could be used as an animal-free NCC system for reproductive toxicity testing.

## Materials and methods

### Clinical description of the enrolled patients with CHARGE syndrome

As shown in *Figure 1—source data 1*, patient1 (CH1), a Japanese male, was born at 39 weeks of gestation with a birth weight of 3.3 kg and a length of 50.5 cm. As major diagnostic criteria, he was noted to have external asymmetrical ear defects and bilateral sensorineural hearing loss (>70 dB). A computed tomography (CT) scan of the temporal bones revealed that semicircular canals were bilaterally hypoplastic. He also showed velopharyngeal incoordination and gastroesophageal reflux. Development was severely delayed, with a developmental quotient of 15 at 3 years old. He had micropenis, cryptorchidism, and delayed incomplete pubertal development. His height was 126.0 cm (– 2.1 s.D.), and his weight was 25.2 kg (– 1.3 s.D.) at 10 years and 6 months old. He was noted to have a distinctive CHARGE physiognomy (*Blake and Prasad, 2006*). By the direct sequencing of his genomic DNA, a heterozygous nonsense mutation in *CHD7*, i.e., c.4171delC p.Gln1391fs*13, was identified. Patient2 (CH2), a Japanese female, was born at 38 weeks of gestation with a birth weight of 3.03 kg and a length of 48.6 cm. She was noted to have external asymmetrical ear defects and bilateral sensorineural hearing loss (>95 dB). A CT scan of the temporal bones revealed that semicircular canals were bilaterally hypoplastic, and the numbers of turns to the cochlea were decreased (Mondini defects). She also showed velopharyngeal incoordination and bilateral retinal coloboma with visual impairment. As minor diagnostic criteria (*Blake and Prasad, 2006*), development was severely delayed with a developmental quotient of 50 at 5 years old. She had delayed incomplete pubertal development. Her height was 119.5 cm (– 2.7 s.D.), and her weight was 21.5 kg (– 1.6 s.D.) at 10 years old. Her physiognomy showed features typical of CHARGE syndrome. By the direct sequencing of her genomic DNA, a heterozygous nonsense mutation in *CHD7*, i.e., c.4480C > T p. Arg1493Ter, was identified.

### Generation and maintenance of iPSCs from dermal fibroblast

As a control, WD39-iPSCs were derived from the HDFs of a healthy 16-year-old Japanese female (*Imaizumi et al., 2012b*). 201B7-iPSCs and WA29-iPSCs were derived from the HDFs of a 36-year-old Caucasian female (Cell Applications Inc., San Diego, CA). 1210B2-iPSCs and 1201C1-iPSCs were derived from human peripheral blood mononuclear cells of a healthy 29-year-old African/American female (Cellular Technology Limited). 201B7-iPSCs, 1210B2-iPSCs, and 1201C1-iPSCs were kindly provided by Shinya Yamanaka. (*Okita et al., 2013*; *Takahashi et al., 2007*) KhES1-ESCs were kindly provided by Norio Nakatsuji (*Suemori et al., 2006*). CH1-iPSCs and CH2-iPSCs were derived from the HDFs of a 10-year-old Japanese male patient and the HDFs of a 10-year-old Japanese female patient, respectively. The clinical diagnoses of these two CHARGE syndrome patients were made based on the Blake criteria (*Blake and Prasad, 2006*). WD39-iPSCs, 201B7-iPSCs, WA29-iPSCs were established through the retroviral transduction of four transcription factors (*KLF4, OCT4, SOX2, and c-MYC*) into HDFs (*Takahashi et al., 2007*), and 1210B2-iPSCs and 1201C1-iPSCs were established using the combination of *KLF4, OCT4, SOX2,L-MYC, LIN28, EBNA* and shRNA for *TP53*, as

previously described (*Okita et al., 2013*). The maintenance of HDFs, stem cell culture, characterization and teratoma formation were performed as described previously (*Imaizumi et al., 2012a2012*; *Ohta et al., 2011*; *Takahashi et al., 2007*). We performed mycoplasma contamination test using MycoAlert Mycoplasma Detection Kits (Lonza Walkersville, Inc., Walkersville, MD) and confirmed all lines were not contaminated by mycoplasma. All human cell and tissue donors were provided explanatory materials and a verbal explanation of the procedure, detailing both the procedure and the purposes of the experiment, as well as their rights, prior to collection and use. All experimental procedures were reviewed and approved by the Keio University School of Medicine Ethics committee (Approval Number: 20080016). RRIDs (Research Resource Identifiers) were provided as below; KhES1 (CVCL_B231), WD39 (CVCL_Y528), 201B7 (CVCL_A324), WA29 (CVCL_LJ40), 1210B2 (CVCL_LJ38), 1201C1 (CVCL_LJ37), CH1#7 (CVCL_LJ32), CH1#11 (CVCL_LJ31), CH1#20 (CVCL_Y955), CK1#25 (CVCL_Y956), CH2#1 (CVCL_LJ#33), CH2#16 (CVCL_LJ#34) and CH2#19 (CVCL_LJ35).

## Teratoma formation assay

To assess the pluripotency of generated iPSCs, we transplanted these iPSCs into the testis of 8-week-old NOD/SCID mice (OYG International) as previously described (*Ohta et al., 2011*). Eight weeks after transplantation, teratomas were dissected and fixed with 4% PFA in PBS. Paraffin-embedded tissue was sectioned and stained with hematoxylin and eosin. Images were obtained with a BZ-9000 (Keyence) microscope. All experimental procedures were reviewed and approved by the Keio University Institutional Animal Care and Use Committee (Approval Number: 09169).

## Sequencing of *CHD7* mutation in enrolled patients

The molecular tests for the *CHD7* gene mutations were conducted as previously reported (*Aramaki et al., 2006a*). We confirmed that the fibroblasts and iPSCs from both CHARGE syndrome patients showed mutations in *CHD7*, whereas the control fibroblasts and iPSCs did not, by sequencing of the PCR amplicons with the primers below using an automated sequencer ABI3100 (Thermo Fisher Scientific, Waltham, MA) as previously described. (*Aramaki et al., 2006a*)

Primer sets:
*CHD7* exon17 F: CTATGCGTCAGGCCTCCTT
*CHD7* exon17 R: TGGGTCTGACTGGTACTCTCTG
*CHD7* exon19 F: TGCAGCATTTGTTTAGTCTGC
*CHD7* exon19 R: TTCCCAATGCATCTTGTAAGC

## qRT-PCR assay

Total RNA was isolated and extracted as previously described. cDNA synthesis from RNA was performed using Superscript III reverse transcriptase (Thermo Fisher Scientific), followed by digestion with RNase H (Thermo Fisher Scientific). qRT-PCR was performed using a 7900HT Real-Time PCR system (Thermo Fisher Scientific) or a Viia7 Real-Time PCR system (Thermo Fisher Scientific) with SYBR green (TaKaRa, Kusatsu, Japan). For every set of qRT-PCR analyses, we had three technical replicates and at least three biological replicates. Data were analyzed by Dunn's multiple comparisons test after Kruskal-Wallis test using GraphPad Prism software version 7.0a (GraphPad Software). The following primers were used:

Primer sets:
*CTGF* F: CAAGGGCCTCTTCTGTGACT
*CTGF* R: ACGTGCACTGGTACTTGCAG
*EDN1* F: GACATCATTTGGGTCAACACTC
*EDN1* R: GGCATCTATTTTCACGGTCTGT
*OLFM3* F: CAGGAGGAAATTGGTGCCTA
*OLFM3* R: AGGGTCTGTCATCCAAGCAC
*POU3F2* F: CGGCGGATCAAACTGGGATTT
*POU3F2* R: TTGCGCTGCGATCTTGTCTAT
TaqMan Gene Expression Assays, Inventoried
*CHD7* primer: Assay ID: Hs00214990_m1
*GAPDH* primer: Assay ID: Hs99999905_m1

## ChIP-qRT-PCR analysis for CHD7

Cells were crosslinked with 1% formaldehyde for 10 min, incubated with 200 mM glycine for 5 min and then stored at −80℃ until use. The ChIP assay was performed as previously described (*Kimura et al., 2008*). Co-immunoprecipitated DNA was used as a template for PCR of the genomic region. The genomic regions were determined by the NCC-specific enhancer regions identified, as previously described (*Rada-Iglesias et al., 2012*). Data were analyzed by paired t test using Graph-Pad Prism software version 7.0a (GraphPad Software). The following primers were used:

Primer sets:
h*POU3F2* distal enhancer F: CAGTAAGCTGCTTGGCCATT
h*POU3F2* distal enhancer R: CAGCCCTCCCTCCTCTTAAC
h*OLFM2* distal; enhancer F: CAATCCCATCTGACCCAACT
h*OLFM2* distal enhancer R: CTGGCTGGTTTCCAGGTTTA
h*EDN1* distal enhancer F: TTCCCTCAGCTTTTGCTTGT
h*EDN1* distal enhancer R: ATTTGGGGGCTTTTTGAGAA
h*CTGF* distal enhancer F: GATTTCAGCTGCTGGCTACC
h*CTGF* distal enhancer R: ATGGCTATCACTTGCCTGCT

## Flow cytometry

Day 10 iPSC-NCCs obtained by Method B were detached with Accutase (Innovative Cell Technologies, San Diego, CA) and collected with ice-cold MACS buffer (Miltenyi Biotec, Bergisch Gladbach, Germany) consisting of phosphate-buffered saline (PBS), 0.5 M EDTA, and 5% bovine serum albumin. After washing, the cells were suspended in ice-cold MACS buffer at $2 \times 10^5$ cells/ml and stained for 30 min at 4℃ using PE-conjugated anti-human CD271 (NGFR) mouse IgG1 antibody (BioLegend, San Diego, CA) and FITC-conjugated anti-human CD57 (B3GAT) mouse IgM antibody (Beckman Coulter, Brea, CA). Propidium iodide staining allowed for the exclusion of dying/dead cells from the analysis. Isotype controls were used as negative controls. Flow cytometric analyses were performed using a FACS Calibur flow cytometer (Becton Dickinson, Franklin Lakes, NJ).

## Immunocytochemical analysis of iPSCs and NCCs

Cells were fixed with PBS containing 4% paraformaldehyde (PFA) for 15 min at room temperature. The cells were analyzed by immunofluorescence staining using the following antibodies: AP2α (monoclonal, 1:100; Cell Signaling Technologies, Danvers, MA), β-III tubulin (monoclonal, 1:1000; Sigma-Aldrich), CD90 (monoclonal, 1:100; BD Pharmingen, San Diego, CA), FOXG1 (polyclonal, 1:250; Abcam, Cambridge, UK), GFAP (monoclonal, 1:200; Thermo Fisher Scientific), Mash1 (monoclonal, 1:500; BD Pharmingen), OTX2 (polyclonal, 1:100; R and D Systems, Minneapolis, MN), P75NTR (polyclonal, 1:500; Abcam), SMA (monoclonal, 1:500; Sigma-Aldrich), SOX10 (polyclonal, 1:200; Abcam), Peripherin (polyclonal, 1:500; Merck Millipore, Billerica, MA), TRA-1–60 (monoclonal, 1:200; Millipore), and TRA-1–81 (monoclonal, 1:200; Merck Millipore). Immunoreactivity was visualized with secondary antibodies conjugated with Alexa 488, Alexa 568, or Alexa 647 (1:1000, Thermo Fisher Scientific). Nuclei were counterstained using Hoechst 33258 (10 µg/ml, Sigma-Aldrich). Images were obtained using an Apotome (Carl Zeiss, Oberkochen, Germany) or LSM-710 confocal (Carl Zeiss) microscope.

## Generation of NCCs by method A

The NCC differentiation of iPSCs was performed as previously described with some modifications (*Lee et al., 2009*; *Lee and Studer, 2010*). Briefly, dissociated iPSCs were plated onto an AggreWell 400 plate (Stem Cell Technologies, Vancouver, Canada) at a density of 600, 000 cells/ well in human ES medium consisting of DMEM/Ham's F12 (Sigma-Aldrich), 20% Knockout Serum Replacement (Thermo Fisher Scientific), 2 mM L-glutamine (Thermo Fisher Scientific), $1 \times 10^{-4}$ M non essential amino acids (Sigma-Aldrich), $1 \times 10^{-4}$ M 2-mercptoethanol (Sigma-Aldrich), and 0.5% penicillin and streptomycin (Thermo Fisher Scientific), the mediun also contained 10 µM Y-27632 (Wako Pure Chemical Industries, Osaka, Japan) in order to make homogenous embryoid bodies (EBs) consisting of 400 cells. After 40 hr, the EBs were transferred to a bacteria dish and cultured in suspension for a week in human EB medium consisting of DMEM/Ham's F12 (Sigma-Aldrich), 5% Knockout Serum Replacement (Thermo Fisher Scientific), 2 mM L-glutamine (Thermo Fisher Scientific), $1 \times 10^{-4}$ M

non-essential amino acids (Sigma-Aldrich), $1 \times 10^{-4}$ M 2-mercaptoethanol (Sigma-Aldrich), 0.5% penicillin and streptomycin, and containing 10 μM SB431542 (R and D Systems), and 250 μg/ml Noggin-Fc (R and D systems). At day 8, the human EB medium was replaced with N2 medium consisting of DMEM/Ham's F12, GlutaMax-I (Thermo Fisher Scientific), 0.5% GlutaMax (Thermo Fisher Scientific), 1% N2 supplement (Thermo Fisher Scientific), 0.5% insulin (Thermo Fisher Scientific), 0.5% penicillin and streptomycin, and containing 10 μM SB431542 (Sigma-Aldrich), and 250 μg/ml Noggin-Fc (R and D Systems). At day 15, the EBs were replaced in a 6-well plate coated with 10 ng/ml fibronectin (Sigma-Aldrich) and cultured in N2 medium supplemented with 20 ng/ml of human recombinant EGF (PeproTech, Rocky Hill, NJ) and 20 ng/ml of human recombinant FGF2 (PeproTech). After 5–7 days of adhesion culture, the cells had migrated out from the colonies were collected and subjected to the analysis. The medium was changed every three days in this protocol.

## Generation of NCCs by method B

iPSCs were differentiated into NCCs, as previously described. (*Bajpai et al., 2010*). Briefly, iPSCs were incubated with 2 mg/ml collagenase IV (Thermo Fisher Scientific). Once the iPSCs were detached, the clusters were broken into pieces consisting of 100–200 cells and plated onto a 100 mm petri dish (Becton Dickinson) in hNCC medium (NC medium). The medium consisted of 1:1 neurobasal medium (Thermo Fisher Scientific) and DMEM/F-12 medium containing 1x GlutaMax (Thermo Fisher Scientific), 5 mg/ml insulin (Sigma-Aldrich), 0.5% penicillin and streptomycin, 0.5x GEM 21 NeuroPlex serum-free supplement (Gemini Bio Products, West Sacramento, CA), 0.5x N2 supplement and supplemented with 20 μg/ml human recombinant EGF and 20 μg/ml human recombinant FGF2. The medium was changed every other day. After seven days of differentiation, migratory NCCs appeared from the attached spheres. At 3–4 days after their appearance, the cells were used for subsequent analysis.

## Multipotency of iPSC-derived NCCs

We induced in vitro differentiation into adipocytes, chondrocytes, and osteocytes as previously reported (*Lee et al., 2010*). Differentiated cells were stained by Toluidine blue (Wako Pure Chemical Industries), Safranin-O (Wako Pure Chemical Industries), and Alizarin red (Wako Pure Chemical Industries). We also differentiated them into myofibroblast (SMA+) and peripheral neurons (peripherin+) and performed an immunocytochemical analysis. In vivo differentiation of iPSC-NCCs into chondrocytes: We dissociated Method B iPSC-NCCs at day 10 with Accutase and purified the TRA-1–60-negative fraction using a MACS system (Miltenyi Biotec). We next injected $1.0 \times 10^6$ TRA-1–60-negative cells into the testes of 8-week-old NOD-SCID mice, as previously described (*Ohta et al., 2011*). Eight weeks after transplantation, the testes were dissected and fixed with 4% PFA in PBS. The paraffin-embedded tissue was sectioned and stained with toluidine blue (performed by Dept. of Pathology, Keio University School of Medicine). Images were obtained using a BZ-9000 (Keyence, Osaka, Japan) microscope.

## Expression array with iPSC-NCCs obtained by method B

Total RNA was isolated from day-10 iPSC-NCCs using TRIzol (Thermo Fisher Scientific) according to the manufacturer's protocol and further purified with an RNeasy mini kit (Qiagen, Hilgen, Germany). Two replicates were run per line from two independent inductions. For the microarray analysis, RNA quality was assessed using a 2100 Bioanalyzer (Agilent Technologies Inc., Santa Clara, CA, USA). Total RNA (100 ng) was reverse-transcribed, biotin-labeled, and hybridized to a Human Genome U133 Plus 2.0 Array (Affymetrix, Santa Clara, CA), which was subsequently washed and stained in a Fluidics Station 450 according to the manufacturer's instructions (*Lockhart et al., 1996*) (*Heishi et al., 2006*). The microarrays were scanned using a GeneChip Scanner 3000 7G (Affymetrix), and the RMA algorithm was implemented for the background correction, normalization across arrays, and log2 transformation of the raw image files (*Bolstad et al., 2003*). Normalized data were filtered based on gene expression level and analyzed using GeneSpring GX software 14.5 (Agilent Technologies) for producing scatter plots and using R package (gplots) for producing heatmaps (*Warnes et al., 2015*). The GeneChip data were deposited in the NCBI Gene Expression Omnibus (GEO; http://www.ncbi.nlm.nih.gov/geo/) and are accessible through the GEO series accession number GSE86212.

## Dispersion assay with Delaunay triangulation

In this assay, we used iPSC-NCCs obtained by Method A. At day 15, EBs replated onto a fibronectin (10 ng/ml)-coated 8well-plastic-bottomed chamber (ASAHI GLASS, Tokyo, Japan) were imaged for 8 hr. To analyze how the cells dispersed from each sphere, the Delaunay triangulation algorithm was used (*Carmona-Fontaine et al., 2011*). ALl cells around the spheres were connected to their closest neighboring cells, and the network shaped triangles by this algorithm. This algorithm is available as an ImageJ plugin. Data were analyzed by Mann-Whitney U test using GraphPad Prism software version 7.0a (GraphPad Software).

## Scattering assay

In this assay, we used iPSC-NCCs obtained by Method A. On day 16, we attached a floating sphere onto the well of a 24-well plate coated with fibronectin (10 µg/ml) and cultured it in N2 medium supplemented with 20 ng/ml of human recombinant EGF and 20 ng/ml of human recombinant FGF2. After five days of adhesion culture, cells were fixed with PBS containing 4% PFA for 15 min at room temperature. F-actin and nuclei were stained using Alexa Fluor-488 phalloidin (Thermo Fisher Scientific) and Hoechst 33258 (Sigma-Aldrich), respectively. Images were obtained using a BZ-9000 (Keyence) microscope. To quantify intercellular contacts of iPSC-NCCs, we analyzed the outermost nine cells in each of the eight 45 degree-sector of a sphere by counting the number of their contacting-neighboring cells (*Figure 4F*), and we classified them into three groups, 0, 1, and >1. Each cell line was analyzed in at least three independent experiments. Data were analyzed Dunnett's multiple comparisons tests after one-way ANOVA or Dunn's multiple comparison test after Kruskal-Walli test using GraphPad Prism software version 7.0a (GraphPad Software).

## xCELLigence assay

We used Method B NCCs for this assay. We dissociated day-10 iPSC-NCCs into single cells with Accutase (Innovative Cell Technologies Inc.). We used the xCELLigence-DP system (Roche) with CIM-Plate 16 to measure the migration index of each type of iPSC-NCCs. The upper plate of CIM-Plate 16 was coated with fibronectin (10 µg/ml in PBS), and 100,000 cells were added to each upper well. NC medium without human recombinant EGF and human recombinant FGF2 was added into each upper well, and NC medium without human recombinant EGF and human recombinant FGF2 containing 10% fetal bovine serum was added to each lower well. Cells that migrated from the upper to the lower well were automatically measured by the xCELLigence system. Eventually, aphidicolin (Sigma, Saint Louis, MO) was used at the concentration of 10 µg/ml. Data were analyzed by Tukey's multiple comparisons test after one-way ANOVA and Sidak's multiple comparisons test after two-way repeated measure ANOVA using GraphPad Prism software version 7.0a (GraphPad Software).

## BrdU incorporation in vitro

Passaged day10 iPSC-NCCs were seeded into wells of an 8-well glass-bottomed plate coated with poly-L-ornithine (0.1 mg/ml) and fibronectin (10 µg/ml) at a low density in NC medium supplemented with 10 µM BrdU (Sigma-Aldrich). After 24 hr, the cells were fixed with PBS containing 4% PFA for 15 min at room temperature and immunostained with sheep polyclonal anti-BrdU antibody (1:500; Fitzgerald Industries International, Acton, MA). Images were randomly captured with an Apotome microscope, and the cells were manually counted. Data were analyzed by unpaired t test using GraphPad Prism software version 7.0a (GraphPad Software).

## Cell adhesion assay

Control iPSC-NCCs and CHARGE iPSC-NCCs obtained by Method B were used for this assay. Each iPSC-NCCs type was cultured to semi-confluence in NC medium, detached by 5 min of treatment with Accutase, and were washed with NC medium twice. We resuspended the cells at a density of 1 × 10⁵ cells per ml in NC medium, and added 100 µl of cell suspension to each well of a fibronectin-coated 96-well plate. After 60 min incubation at 37°C, we changed the medium and added 10 µl of WTS-1/ECS (MerckMillipore, Billerica, USA) per well except for 12 wells per 96-well plate. After 90 min of incubation at 37°C, the plate was shaken thoroughly for 1 min on a shaker, and then the absorbance at 450 nm of the treated and untreated samples was measured using a microplate

reader. The average of absorbance values of the 12 wells without WTS-1/ECS was considered a baseline, and the data were normalized to that of 201B7 iPSC-NCCs in each experiment. Data were analyzed by Dunnett's multiple comparisons test after one-way repeated measures ANOVA using GraphPad Prism software version 7.0a (GraphPad Software). (*Chen et al., 2009*) (*Mobley and Shimizu, 2001*).

## Spontaneous single cell motility assay

Control iPSC-NCCs and CHARGE iPSC-NCCs obtained by Method B were used for this assay. At day 10–12, after the beginning of differentiation by Method B, the remaining spheres were removed by direct aspiration with a fine Pasteur pipette. The adherent NCCs were washed 2–3 times gently with PBS, and then stained for 3 hr at 37°C with either Vybrant DiI (Thermo Fisher Scientific) or Vybrant DiO (Thermo Fisher Scientific) diluted 1/300 in NC medium. Notably, permutations of the staining dyes confirmed that the nature of the dye had no effect on the migratory behavior of the cells. After 4–5 washes with PBS, the stained NCCs were then dissociated using Accutase, counted using Trypan Blue (Wako Pure Chemical Industries), and then co-seeded in equal amounts at a density of $5 \times 10^3$ cells/well (total of $10 \times 10^3$ cells per well) onto 8-well, plastic-bottomed chambers that were previously coated with fibronectin at 10 µg/ml. Three hours after seeding, once the cells had attached, the chambers were transferred to an LSM 5, PASCAL Exciter confocal microscope (Carl Zeiss) that was equipped with a heat- (37°C) and gas-controlled incubation chamber (5% $CO_2$) (Tokai Hit, Shizuoka, Japan) that was coupled to a heated motorized stage. The objective lens (EC-Plan Neofluar, 10 X, Numerical Aperture 0.3) was maintained at 37°C and was used to acquire a Z-stack time-lapse series (7 Z-stacks spanning 30 µm, every 15 min) of multiple locations. Z-projections were produced using Image Browser Zeiss software at the end of the analysis. The time-lapse recordings began at 4 hr after seeding and continued for at least 16 hr. Individual cells were manually tracked using the Manual Tracking plugin of the Fiji software (1.48). Cells exhibiting abnormal morphologies (e.g., neurite-like, or with signs of apoptosis) were excluded from the analysis. Calculations of individual velocities and directionalities were performed using the chemotaxis and migration tool from Ibidi (Martinsried, Germany). Data were analyzed Sidak's multiple comparisons tests after two-way repeated measures ANOVA using GraphPad Prism software version 7.0a (GraphPad Software).

## In ovo experiments

Control and CHARGE floating spheres at day seven were stained with Vybrant DiI and Vybrant DiO respectively for 6 hr each and were seeded into the same 100 mm Petri dish to make mixture of dual colored NCCs sheet. On day 10 dual-colored NCCs were dissected with a needle as a cluster and transplanted into the dorsal side (top) of the developing neural tube at the hindbrain level of HH stage 8–10 chick embryos. The embryos were incubated for 36 hr, and imaged with an SVZ16 (Olympus, Tokyo, Japan) stereo microscope.

## Time-lapse imaging of transplanted cells during in ovo experiments

We transplanted control and CHARGE iPSC-NCCs into chick embryos, as described above. At 6 hr after transplantation, we started to perform time-lapse imaging as previously described (*Tabata and Nakajima, 2003*). Briefly, transplanted chick embryos were placed on a Millicell-CM membrane (pore size, 0.4 µm; Millipore) and cultured in saline, which is described below. The dishes were then mounted onto a confocal microscope (FV1000, Olympus Optical). Approximately, 20 optical Z-section images were acquired at an interval of 5 µm every 15 min, and all focal planes (100 µm) were merged. Individual cells were manually tracked using the Manual Tracking plugin of the Fiji software (1.48). Data were analyzed by two-way repeated measures ANOVA using GraphPad Prism software version 7.0a (GraphPad Software).

The saline used consisted of the following: solution A (for 1 l): 121.0 g of NaCl, 15.5 g of KCl, 10.4 g of $CaCl_2.2H_2O$, and 12.7 g of $MgCl_2.6H_2O$; solution B (for 1 l): 2.4 g of $Na_2HPO_4.2H_2O$ and 0.2 g of $NaH_2PO_4.2H_2O$. After autoclaving but prior to using the solutions, mix 120 ml of solution A with 2700 ml of $H_2O$; then, add 180 ml of solution B, as previously described (*Psychoyos and Finnell, 2008*).

## Acknowledgements

We would like to thank Professors Shinya Yamanaka (CiRA, Kyoto University) and Norio Nakatsuji (Kyoto University) for providing 201B7 iPSCs and KhES1 cells, respectively. We are grateful to Yu Yamaguchi for technical assistance and suggestions, and to all members of the Okano laboratory for their encouragement and support. We thank Douglas Sipp (Keio University) for invaluable comments regarding the manuscript. This work was supported by funding from the Project for the Realization of Regenerative Medicine; Support for the Core Institutes for iPS Cell Research from the Ministry of Education, Culture, Sports, Science and Technology of Japan (MEXT; to H Okano); and a Grant-in-Aid for the Global COE Program from MEXT to Keio University. This work was also supported by a Grant-in-Aid for Young Scientists (B) from MEXT (project number: 26860823), a Keio University Grant-in-Aid for the Encouragement of Young Medical Scientists to H Okuno., and by a Grant-in-Aid for Scientific Research on Innovative Areas from MEXT to K Nakajima (project number: JP16H06482). H Okano is a scientific consultant for SanBio,Co. Ltd. and K Pharma Inc.

## Additional information

### Competing interests

Hideyuki Okano: H Okano is a scientific consultant for San Bio, Co. Ltd., Esai, Co. Ltd., and Dainichi Sankyo, Co. Ltd. The other authors declare that no competing interests exist.

### Funding

| Funder | Grant reference number | Author |
| --- | --- | --- |
| Keio University School of Medicine | Grant-in-Aid for the Encouragement of Young Medical Scientists | Hironobu Okuno |
| Ministry of Education, Culture, Sports, Science, and Technology | Project for Realization of Regenerative Medicine and Support for Core Institutes for iPSC research | Hideyuki Okano |
| Ministry of Education, Culture, Sports, Science, and Technology | A Grant -in-Aid for the Global COE program | Hironobu Okuno Hideyuki Okano |
| Ministry of Education, Culture, Sports, Science, and Technology | A Grant-in-Aid for Young Scientists (B) | Hironobu Okuno |
| Ministry of Education, Culture, Sports, Science, and Technology | A Grant-in-Aid for Scientifc Reserch on Innovative Areas | Kazunori Nakajima |
| Ministry of Education, Culture, Sports, Science, and Technology | Support for Core Institutes fro iPS Cell Research | Hideyuki Okano |

The funders had no role in study design, data collection and interpretation, or the decision to submit the work for publication.

### Author contributions

Hironobu Okuno, Conceptualization, Resources, Data curation, Software, Formal analysis, Supervision, Funding acquisition, Validation, Investigation, Visualization, Methodology, Writing—original draft, Project administration, Writing—review and editing; Francois Renault Mihara, Conceptualization, Data curation, Formal analysis, Supervision, Validation, Investigation, Methodology, Writing—original draft, Writing—review and editing; Shigeki Ohta, Conceptualization, Formal analysis, Supervision, Validation, Investigation, Methodology, Writing—original draft; Kimiko Fukuda, Data curation, Formal analysis, Supervision, Investigation, Visualization, Methodology, Writing—original draft; Kenji Kurosawa, Conceptualization, Resources, Data curation, Investigation, Writing—original draft; Wado Akamatsu, Conceptualization, Resources, Supervision, Investigation, Writing—original draft; Tsukasa

Sanosaka, Conceptualization, Data curation, Software, Formal analysis, Investigation, Visualization, Methodology, Writing—original draft; Jun Kohyama, Formal analysis, Supervision, Validation, Investigation, Methodology, Writing—original draft, Writing—review and editing; Kanehiro Hayashi, Formal analysis, Supervision, Investigation, Visualization, Methodology, Writing—review and editing; Kazunori Nakajima, Software, Supervision, Funding acquisition, Visualization, Writing—review and editing; Takao Takahashi, Conceptualization, Resources, Supervision, Investigation, Writing—original draft, Project administration; Joanna Wysocka, Conceptualization, Supervision, Investigation, Methodology, Writing—original draft, Writing—review and editing; Kenjiro Kosaki, Conceptualization, Resources, Supervision, Funding acquisition, Investigation, Methodology, Writing—original draft, Writing—review and editing; Hideyuki Okano, Conceptualization, Supervision, Funding acquisition, Investigation, Writing—original draft, Project administration, Writing—review and editing

#### Author ORCIDs
Hironobu Okuno (iD) http://orcid.org/0000-0003-1932-9482
Kazunori Nakajima (iD) https://orcid.org/0000-0003-1864-9425
Hideyuki Okano (iD) https://orcid.org/0000-0001-7482-5935

#### Ethics
Human subjects: All human cell and tissue donors were provided explanatory materials and a verbal explanation of the procedure, detailing both the procedure and the purposes of the experiment, as well as their rights, prior to collection and use. All experimental procedures were reviewed and approved by the Keio University School of Medicine Ethics committee (Approval Number: 20080016).
Animal experimentation: All experimental procedures were reviewed and approved by the Keio University Institutional Animal Care and Use Committee (Approval Number: 09169).

#### Decision letter and Author response
Decision letter https://doi.org/10.7554/eLife.21114.033
Author response https://doi.org/10.7554/eLife.21114.034

## Additional files

### Supplementary files
• Transparent reporting form
DOI: https://doi.org/10.7554/eLife.21114.028

### Major datasets
The following dataset was generated:

| Author(s) | Year | Dataset title | Dataset URL | Database, license, and accessibility information |
|---|---|---|---|---|
| Okuno H | 2016 | CHARGE syndrome modeling using patient-derived iPSC reveals defective migration of neural crest cells harboring CHD7 mutations | https://www.ncbi.nlm.nih.gov/geo/query/acc.cgi?acc=GSE86212 | Publicly available at the NCBI Gene Expression Omnibus (accession no. GSE86212) |

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
