## [Decision Letter]

Thank you for submitting your article "CHARGE syndrome modeling using patient-iPSCs reveals defective migration of neural crest cells harboring CHD7 mutations" for consideration by *eLife*. Your article has been reviewed by two peer reviewers, and the evaluation has been overseen by Marianne Bronner as the Senior and Reviewing Editor. The reviewers have decided to remain anonymous.

The reviewers have discussed the reviews with one another and the Reviewing Editor has drafted this decision to help you prepare a revised submission.

Summary:

This is a well-conducted and relevant study modeling the CHARGE syndrome with induced pluripotent stem cells. The results suggest for the first time an exciting functional phenotype of neural crest cells from CHARGE patients. They observed a migratory deficit consistent with the patient phenotype making this study a very promising line of work. The reviewers agree that this study is potentially interesting and appropriate for *eLife*. However, the paper needs additional analyses to study cell migration as well as better controls and numbers to guarantee significance.

Essential revisions:

1) The characterization of the migratory/dispersion phenotype shown in Figure 4 is not convincing. The image quality in Figure 4 is inadequate. The authors only show a movie of control cells, but none of CHARGE cells which would be needed to evaluate the CHARGE phenotype.

2) Analysis of dispersion needs to be improved. The ratio of cell number with non-attached neighboring cells is not appropriate for quantification as it depends on the number of cells analyzed, the perimeter of the cell cluster, etc. Distance to the closer neighbor expressed as a ratio between t0 and tn, using Delaunay triangulation would be a better measure. They should also analyze other parameters like cell speed, persistence, cell area, time of contact etc.

3) The trans-well assay employed to study cell motility is not ideal, as it is prone to several artifacts (e.g. differential adhesion, differential proliferation, etc.). Time-lapse imaging of individually migrating cells is the most direct assay for cell motility. This analysis is shown in Figure 6. But why is there a continuous increase in cell velocity? This is an important observation that needs to be investigated further. Cell velocity is a cell property that can be affected by external stimulus of by cellular activities; therefore if the external and internal conditions of a cell are stable, the velocity should be constant. If the conditions are changing, it won't be possible to compare two cell populations (control and CHARGE cells).

4) The transplantation experiments to study cell migrating in vivo are interesting, but the analysis is not adequately performed. Time lapse movies and analysis of cell motility in vivo should be performed.

5) The authors show one control and two CHARGE syndrome iPSC lines but these are not isogenic. Can they rule out possible allelic variations? To address this, the authors should either increase the numbers of control and CHARGE lines to a minimum of 3 lines each to be statistically conclusive or prepare an isogenic control.

6) The global gene expression studies are convincing and provide some discussion on possible involvement of altered genes and pathways. However no further verification/ confirmation has been made. This would greatly enhance this paper.

7) To corroborate the migratory phenotype, they should include further analysis of intracellular filaments/focal adhesion/integrin machinery to suggest molecular culprits of the migratory phenotype.

---

## [Author Response]

Essential revisions:1) The characterization of the migratory/dispersion phenotype shown in Figure 4 is not convincing. The image quality in Figure 4 is inadequate. The authors only show a movie of control cells, but none of CHARGE cells which would be needed to evaluate the CHARGE phenotype.

Per the suggestion of the comment, we replaced the image of the

migratory/dispersion phenotype to better represent the iPSCs-NCCs in Figure 4 (Figure 4, and 4E). We have added a movie of CHARGE cells in Figure 4—video 2, in addition to a movie of control cells in Figure 4—video 1.

2) Analysis of dispersion needs to be improved. The ratio of cell number with non-attached neighboring cells is not appropriate for quantification as it depends on the number of cells analyzed, the perimeter of the cell cluster, etc. Distance to the closer neighbor expressed as a ratio between t0 and tn, using Delaunay triangulation would be a better measure. They should also analyze other parameters like cell speed, persistence, cell area, time of contact etc.

a) Analysis of dispersion using Delaunay triangulation

Per the suggestion of the reviewer comment, we analyzed the dispersion of migrating cells from a sphere using the Delaunay triangulation algorithm, and we could easily recognize the difference between the control and CHARGE cells using the algorithm. The statistical analysis of increased triangle area between t=0 and t=8h in the control and CHARGE cells revealed a delayed dispersion of cells in CHARGE-spheres. We have added the results of the assay to Figure 4, and 4D. We have added the following sentences to the Results section of the revised manuscript to explain the additional dispersion assay using a Delaunay triangulation:

“…neighbors (Figure 4). To clarify this difference, we performed a time-lapse analysis of the initial phase of cell dispersion from the sphere. […] Moreover, to determine whether CHARGE iPSC-NCCs have defects in premigratory-to-migratory transition, we analyzed the intercellular contacts of both control and CHARGE iPSC-NCCs at Phase 2. We visualized…”

b) Analysis of cell speed, persistence, cell area, time of contact.

We apologize for not showing a video of CHARGE cells in our previous submission. Without the video, it is difficult to understand how CHARGE cells migrate out from a sphere. As shown in Figure 4—video 1 and Figure 4—video 2, at first, cells began to spread out as a continuous monolayer (Phase 1); then, cells at the outermost periphery began to disperse (Phase 2). During Phase 1, cell speed seemed to be dependent on the spreading speed of each sphere, which was especially clear for CHARGE cells. The spreading speed was quite different from sphere to sphere. As such, it was difficult to analyze the spreading speed in this assay. Instead, in the analysis shown in Figure 6 of the time-lapse imaging of individually migrating cells, control and CHARGE cells were observed under the same condition, and by measuring the velocity of each cell type, we found defective motility in CHARGE iPSC-NCCs in the present revised manuscript.

Regarding cell persistence, or directionality, cells just moved to the outside of a sphere along with sphere spreading during Phase 1, so directionality was also along the direction of sphere spreading. During Phase 2, as cells at the outermost periphery began to migrate apart, it was a good phase for measuring directionality. However, when CHARGE cells started to migrate away from the spheres, the migrating cells began to leave the observation field. In the analysis shown in Figure 6 of the time-lapse imaging of individually migrating control and CHARGE cells under the same condition (in co-culture), we measured the directionality of the cells and found no difference between the control and CHARGE iPSC-NCCs.

Regarding the time of contact, in the beginning of this assay, cells contacted

each other because they were all from a sphere. In this assay, we focused on observing how cells spread out from a sphere and began to disperse. When they started to migrate apart from each other (Phase 2), especially for CHARGE cells, many cells left the observation field. Along with the analysis of the scattering assay, shown in Figure 4, we could reveal significant persistent intercellular contacts among CHARGE iPSC-NCCs compared with control iPSC-NCCs.3) The trans-well assay employed to study cell motility is not ideal, as it is prone to several artifacts (e.g. differential adhesion, differential proliferation, etc.). Time-lapse imaging of individually migrating cells is the most direct assay for cell motility. This analysis is shown in Figure 6. But why is there a continuous increase in cell velocity? This is an important observation that needs to be investigated further. Cell velocity is a cell property that can be affected by external stimulus of by cellular activities; therefore if the external and internal conditions of a cell are stable, the velocity should be constant. If the conditions are changing, it won't be possible to compare two cell populations (control and CHARGE cells).

1) The artifacts of differential adhesion and differential proliferation in the transwell assay.

Per the suggestion of the reviewer comment, we performed adhesion and proliferation assays to determine whether there was a difference between control and CHARGE iPSC-NCCs grown onto fibronectin (the extracellular matrix protein we used in the transwell assay).

Regarding the effect of differential adhesion, we performed a cell adhesion assay to fibronectin and then found no significant difference between the control and CHARGE iPSC-NCCs (Figure 5—figure supplement 1). Regarding the effect of differential proliferation, we first noted that use of the anti-mitotic drug aphidicolin, an inhibitor of DNA replication, did not significantly affect the migration rate of NCCs in transwell assay, indicating that proliferation is not significantly contributing to the increased impedance over time, as shown in Figure 5—figure supplement 1. We also performed a BrdU incorporation assay with control and CHARGE iPSC-NCCs, and no significant difference was observed, also shown in Figure 5—figure supplement 1. By these assays, we could conclude that the reduction in the number of migrating CHARGE iPSC-NCCs observed in the transwell assay was not due to reduced proliferation and lower CHARGE iPSC-NCC adherence to fibronectin.

We consider that the transwell assay with the xCELLigence system is a good tool for monitoring the continuous kinetic record of cell migration and quantifying the defective migration in CHARGE iPSC-NCCs objectively. These results are also supported by the defective cell motility of CHARGE iPSC-NCCs revealed by the single-cell motility analysis, as shown in Figure 6. We have added the following sentences to the Results section of the present revised manuscript:

“…cells (Figure 5). To exclude the possibility that the reduction in the number of migrating CHARGE iPSC-NCCs was due to reduced proliferation, we treated control iPSC-NCCs with an antimitotic, aphidicolin. […] Taken together, these results suggest that CHARGE iPSC-NCCs exhibit aberrant migration, in contrast with their preserved capacity for proliferation and adherence to fibronectin.”

2) A continuous increase in cell velocity in the time-lapse imaging of individually migrating cells.

As mentioned by the reviewer comment, we have observed a continuous increase in cell velocity in both control and CHARGE iPSC-NCCs. We are not sure what causes the continuous increase in cell velocity, but the phenomenon has been observed in multiple independent experiments. This effect (progressive increase of velocity over time) has not been reported in the literature regarding neural crest cells. Recently it was reported that a continuous increase in cell velocity of metastatic human sarcoma and carcinoma cells was observed in vitro as the cell density increases (Jayatilaka et al., 2017). Even though we didn't observe such cell density increase during the assay, there is some possibility that autocrine/paracrine signaling may cause a continuous increase in cell velocity. Although the reason is unknown, it should be emphasized that the control and CHARGE iPSC-NCCs were recorded while in co-culture so that they encountered the same condition. We explicitly stated that the velocity increased over time, and have added the following sentences to the Results and Discussion sections of the present revised manuscript:

“…iPSC-NCCs. To exclude cell-density effects, we analyzed the motility of mixed iPSC-NCCs, i.e., control + CHARGE, within the same well (Figure 6). The tracking of individual iPSC-NCCs revealed that the average velocities progressively increased over the course of the recording period (Figure 6).

Notably,..”

“Third, a spontaneous motility assay allowed us to assess whether defective motility is a partial cause of the defective migration of CHARGE iPSC-NCCs (Figure 6). […] The observed spontaneous defective motility of the CHARGE cells suggests that such soluble factors are not involved in the defective migration of CHARGE iPSC-NCCs.”

* Jayatilaka, H., Tyle, P., Chen, J. J., Kwak, M., Ju, J., Kim, H. J.,. . . Wirtz, D. (2017). Synergistic IL-6 and IL-8 paracrine signalling pathway infers a strategy to inhibit tumour cell migration. Nat Commun, 8, 15584. doi:10.1038/ncomms15584

4) The transplantation experiments to study cell migrating in vivo are interesting, but the analysis is not adequately performed. Time lapse movies and analysis of cell motility in vivo should be performed.

Per the suggestion of the reviewer comments, we have added a representative time-lapse movie as an illustration of the defective migration of CHARGE iPSC-NCCs transplanted in ovo as shown in Figure 7—video 1, and a quantitative analysis of the data as shown in Figure 7—figure supplement 1.

We have repeatedly performed transplantation of a mixture of control and CHARGE iPSC-NCCs into a chick embryo, but only few embryos were survived. Besides, due to the movements of the developing chick embryos, capturing and tracking the implanted cells over a significant period of time after the initial phase were revealed to be extremely difficult. Therefore, we limited the observation time-window to the first 6 hours after transplanted cells started to migrate. We provided a representative time-lapse movie as an illustration of the defective migration of CHARGE iPSC-NCCs transplanted in ovo compared with control cells. We manually tracked individual migratory control and CHARGE cells in an embryo over time using ImageJ, and revealed significant reduced cell velocity in CHARGE in ovo using total nine embryos by two-way ANOVA (time, and cell type). Along with the analysis of cell motility in vivo with time-lapse movies, we think that our scoring system shown in Figure 7 reflects faithfully the biological events: while the control NCCs exhibited superior behavior compared with the CHARGE NCCs in some embryos, there were also some embryos where both cell types behaved similarly. Whether this variability is a reflection of the extent of a malformation observed in CHARGE patients is unknown.

We have added a representative time-lapse video to Figure 7—video 1, the following images to Figure 7—figure supplement 1, and the following sentences to the Results section of the revised manuscript:

“…transplantation (Figure 7).First, to examine the serial migration of the transplanted cells, we transferred the transplanted embryo to a glass-bottomed plate (IWAKI) 6 h after transplantation and then acquired time-lapse images every 20 min (Figure 7—video 1). […] (Figure 7—figure supplement 1—source data 1, Tab1) Collectively, CHARGE iPSC-NCCs exhibited a lower velocity compared with that of the co-transplanted control iPSC-NCCs (P = 0.03; Wilcoxon signed-rank test) (Figure 7—figure supplement 1). Second,..”

5) The authors show one control and two CHARGE syndrome iPSC lines but these are not isogenic. Can they rule out possible allelic variations? To address this, the authors should either increase the numbers of control and CHARGE lines to a minimum of 3 lines each to be statistically conclusive or prepare an isogenic control.

Per the suggestion of the comment, we added several lines to the control and CHARGE groups to meet the minimum of 3 lines per group for obtaining statistically conclusive evidence of the defective migration of CHARGE iPSC-NCCs. To perform the following functional migration assay, as shown in Figure 4, Figure 5, Figure 6, and 7, we had to prepare both control and CHARGE iPSC-NCCs at the same time. Sometimes we failed to induce a sufficient number of iPSC-NCCs to analyze, so we always maintained several iPSC lines and differentiated NCCs from them. As such, the line # combinations are not the same for each migration analysis. However, we obtained similar results from each cell line combination, and we think that these results accurately reflect the biological events. We have revised Figure 1, Figure 4, and 5C to reflect the line numbers in each group and have also made a new figure or a new source data to be representative of the number of each line in the assay, as shown in Figure 4, Figure 7—figure supplement 1 and Figure 7—figure supplement 1—source data 1.

Figure 1: We used 3 lines (WD39, 201B7, 1210B2) as control cells and 36 lines (CH1#1, CH1#3-CH1#21, CH1#24, CH1#25, CH2#1-CH2#3, CH2#5, CH2#7, CH2#8, CH2#10, CH2#16-CH2#23) as CHARGE cells.

Figure 4 (Dispersion assay): We used 3 lines (WD39, 201B7, 1201C1) as control cells and 3 lines (CH1#20, CH1#25, CH2#1) as CHARGE cells.

Figure 4 (Scattering assay): We used 4 lines (KhES1, WA29, 201B7, WD39) as control cells and 5 lines (CH1#20, CH1#25, CH2#1, CH2#16, CH2#19) as CHARGE cells.

Figure 5 (transwell migration assay): We used 3 lines (WD39, 201B7, 1210B2) as control cells, 3 lines (CH1#7, CH1#20, CH1#25) from CH1, and 3 lines (CH2#1, CH2#16, CH2#19) from CH2.

Figure 6 (spontaneous motility assay): We used 2 lines (WD39, 201B7) as control cells and 2 lines (CH1#25, CH2#16) as CHARGE cells. In this assay, to maintain the control and CHARGE iPSC-NCCs under identical conditions, we compared recordings of co-cultured control and CHARGE iPSC-NCCs. Thus, the statistical analysis was also performed using 1 control line and 1 CHARGE line in each experiment. The defective motility of the CHARGE iPSC-NCCS observed in this assay is compatible with the results from the transwell migration assay, as shown in Figure 5. The transwell migration assay was performed with 4 control lines and 5 CHARGE lines.

Figure 7—figure supplement 1 and Figure 7—figure supplement 1—source data 1 (in vivo migration assay): We used 3 control lines (WD39, 201B7, 1201C1) and 3 CHARGE lines (CH1#25, CH2#1, CH2#16). As with Figure 6, we compared control and CHARGE iPSC-NCCs co-transplanted in an embryo to maintain the control and CHARGE iPSC-NCCs under identical conditions. Thus, the statistical analysis was also performed using 1 control line and 1 CHARGE line in each experiment.

6) The global gene expression studies are convincing and provide some discussion on possible involvement of altered genes and pathways. However no further verification/ confirmation has been made. This would greatly enhance this paper.

Per the suggestion of the comment, we additionally performed ChIP-qPCR for *CHD7* with control iPSC-NCCs to examine whether the potential targets identified in our study are directly regulated by CHD7 binding. The genomic regions were determined by the NCC-specific enhancer regions identified using the same NCC differentiation method we used in this paper (Method B), as previously described (Rada-Iglesias et al., 2012). As shown in Figure 3—figure supplement 1, we found direct CHD7 binding to the distal enhancer sites of *EDN1*. We have added the following sentences to the Results section and Discussion section of the revised manuscript:

“…the analysis (Figure 3—figure supplement 1). These genes are also listed as CHD7 targets in the dataset. […] The results of these transcriptome analyses support the notion that NCCs exhibit migratory and/or cell adhesion defects during embryonic development in CHARGE patients.”

“…dataset (Rouillard et al., 2016). Although these target sites vary depending on cell type, we found the target site of CHD7 in the EDN1 distal promoter region by ChIP-qPCR for CHD7 using our cells. This result suggests that CHD7 regulates not only the expression of some specific key genes but also the robust gene expression in early NCCs.”

7) To corroborate the migratory phenotype, they should include further analysis of intracellular filaments/focal adhesion/integrin machinery to suggest molecular culprits of the migratory phenotype.

Per the suggestion of the comment, we have performed immunostaining for key regulators of cell shape and dynamics: β-actin, F-actin (stained by phalloidin), and α- tubulin. However, we couldn’t see any difference between control and CHARGE iPSC-NCCs regarding either the intensity of immunoreactivities, or the patterns of staining (see Author response image 1). It thus seems necessary to perform specific assays to reveal subtle differences between control and CHARGE-NCCs regarding the mechanism involved in the defective migration of CHARGE-NCCs.

In line with this idea, we have thereafter examined the cell surface expression of integrins alpha4 and beta1, which are especially relevant to interactions with fibronectin. FACS analysis has revealed a tendency to a reduction in surface intensities of both integrins alpha4 and beta1 (see Author response image 2). Strachan et al. have shown that these proteins are particularly important for the migration of cranial NCCs onto fibronectin (Strachan and Condic, 2003). However, consistent data from the literature suggests that the possibly reduced expression of these integrins would positively affect the velocities of NCCs migration at the relatively high concentrations of fibronectin (10 µg/ml) used in our migration assays (transwell and spontaneous single cell motility). On the contrary, CHARGE iPSC-NCCs exhibited reduced migration capabilities compared to control iPSC-NCCs in these conditions, indicating that the modulation of surface integrin expression is not likely to be involved in the defective migration of CHARGE NCCs.

According to such a situation, we did not include these results in the manuscript.

**Author response image 2. respfig2:** The results of flow cytometric analysis of control and CHARGE iPSC-NCCs from method-B (control; red line, CHARGE; blue line). Isotype controls were used as negative controls (black line).

While the differences of immunolocalization of these molecules are not obvious, we could detect that the expression levels of various genes involved in cell adhesion and migration are changed in CHARGE cells by transcriptome analysis (Figure 3). Thus, expression of these molecules is likely to be orchestrated by CHD7 and would result in the remarkable phenotypes in CHARGE cells in our model.

* Strachan, L. R., & Condic, M. L. (2003). Neural crest motility and integrin regulation are distinct in cranial and trunk populations. Developmental Biology, 259(2), 288-302.[2]